# QuITE: Query-Based Irregular Time Series Embedding

**Junghoon Lim** [1]

## Abstract

Irregular Multivariate Time Series (IMTS) are common in practice, yet their irregular sampling complicates effective modeling. Existing approaches typically either (i) design specialized architectures that limit the reuse of proven Multivariate Time Series (MTS) models, or (ii) map IMTS onto regular temporal grids through interpolation, which may distort temporal dynamics by introducing artificial values. To address these limitations, we propose a new *input-embedding-based approach*. We identify that the key bottleneck lies not in the backbone architecture, but in conventional embedding layers that assume uniform sampling. In this work, we introduce **QuITE** (**Qu**ery-Based **I**rregular **T**ime Series **E**mbedding), a simple yet effective plug-and-play embedding module for IMTS. QuITE employs learnable query tokens to aggregate irregular observations through a single self-attention layer, directly producing backbone-compatible latent representations without artificial value generation or architectural modification. Extensive experiments on real-world benchmarks show that QuITE consistently improves MTS models, yielding average relative gains of up to $54.7\%$ in forecasting and $15.8\%$ in classification across diverse datasets and backbone architectures. Code is available at: https://github.com/Meaningfull9502/QuITE.

## 1 Introduction

Irregular multivariate time series (IMTS) arise naturally in many real-world domains, including healthcare, industrial monitoring, and climatology (Zhang et al., 2022; Chowdhury et al., 2023; Chang et al., 2026). IMTS are characterized by non-uniform observation intervals and asynchronous

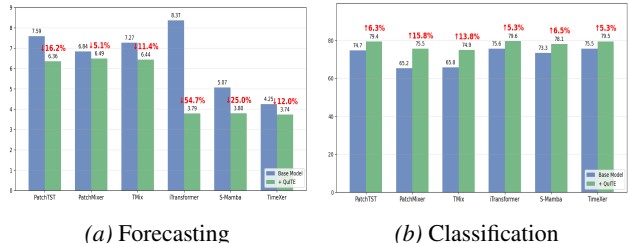

*(a)* Forecasting          *(b)* Classification

*Figure 1.* **Effectiveness of QuITE.** QuITE consistently improves performance across diverse datasets and backbone architectures. Values indicate the average performance over all datasets.

measurements across variables, which pose significant challenges for effective modeling.

Existing methods for handling IMTS can be broadly categorized into two groups: (i) *architecture-based approaches*, which design specialized architectures for IMTS, and (ii) *data-based approaches*, which map IMTS onto regular temporal grids through interpolation at either the raw-data or representation level. Although these approaches have shown partial success, both involve inherent trade-offs. Architecture-based approaches may not fully leverage well-established multivariate time series (MTS) models, despite their extensive validation and proven effectiveness. Data-based approaches, on the other hand, allow MTS models to be reused without architectural modification, but introduce artificial values that may distort the underlying temporal dynamics and lead to suboptimal performance (Luo et al., 2025; Chang et al., 2026).

To address these limitations, we propose a new approach based on *input-embedding*. Our key insight is that the primary bottleneck lies not in the backbone architecture itself, but in conventional input embedding layers, which typically assume uniformly sampled inputs and are therefore ill-suited for IMTS. By handling irregularity at the input embedding stage, state-of-the-art (SOTA) MTS models can be effectively adapted to IMTS, allowing them to better leverage their modeling capacity without architectural changes or artificial value generation.

However, designing such an embedding is non-trivial. Simple fusion strategies, such as adding time embeddings to value embeddings or concatenating values and timestamps, remain limited by conventional embedding schemes designed for uniformly sampled inputs. Although attention

---

[1]SK Shieldus, Seongnam, Republic of Korea. Correspondence to: Junghoon Lim <junghoon9502@gmail.com>.

*Proceedings of the $43^{rd}$ International Conference on Machine Learning*, Seoul, South Korea. PMLR 306, 2026. Copyright 2026 by the author(s).

| Approach | No Artificial Value | Model Flexibility |
|---|:---:|:---:|
| Architecture-based | ✓ | ✗ |
| Data-based | ✗ | ✓ |
| Input-embedding-based (Ours) | ✓ | ✓ |

*Table 1.* Comparison of Different Approaches for IMTS Modeling.

mechanisms can capture interactions among irregular observations, their observation-level outputs require additional pooling to match the structured inputs expected by modern MTS models, such as variable- or patch-level representations. This pooling may dilute fine-grained temporal information.

In this work, we introduce **QuITE** (**Qu**ery-Based **I**rregular **T**ime Series **E**mbedding), a simple yet effective plug-and-play embedding module for IMTS. QuITE employs learnable query tokens as structured aggregation anchors, directly transforming irregular observations into backbone-compatible embeddings. Through a single self-attention layer, these query tokens aggregate information from irregular observations to produce fixed-dimensional latent representations. By bypassing lossy pooling and artificial value generation, the resulting embeddings can be directly fed into existing MTS models without architectural modification. While conceptually inspired by the `[CLS]` token in BERT (Devlin et al., 2019), QuITE repurposes learnable query tokens as an input embedding mechanism for IMTS.

We conduct extensive experiments on multiple real-world IMTS benchmarks to evaluate the effectiveness of the proposed method. The results show that integrating QuITE into existing MTS models consistently improves performance, achieving average relative gains of up to $54.7\%$ in forecasting and $15.8\%$ in classification across diverse datasets and backbone architectures.

Our main contributions are summarized as follows:

- We present a new approach for IMTS modeling at the *input-embedding level*, rather than through architectural redesign or data interpolation.

- We propose **QuITE** (**Qu**ery-Based **I**rregular **T**ime Series **E**mbedding), a simple yet effective plug-and-play embedding module for IMTS. QuITE introduces a set of learnable query tokens that aggregate irregular observations via self-attention.

- Extensive evaluations on diverse IMTS benchmarks demonstrate that QuITE consistently improves MTS backbones, with average relative gains of up to $54.7\%$ in forecasting and $15.8\%$ in classification.

## 2  Related Work

### 2.1  Multivariate Time Series Modeling

Recent MTS forecasting models can be broadly categorized into patch-based or variable-based approaches. PatchTST (Nie et al., 2022) first introduces patch-wise modeling to enhance temporal locality, followed by Pathformer (Chen et al., 2024) with multi-scale patches and PatchMixer (Gong et al., 2023) with CNN-based patch mixing for modeling intra- and inter-variable dependencies. In contrast, variable-based models explicitly focus on cross-variable correlations. iTransformer (Liu et al., 2023) first treats each variate as a token, enabling attention-based modeling of multivariate dependencies, while S-Mamba (Wang et al., 2025) captures inter-variable relationships using a linear-complexity Mamba architecture. To leverage the strengths of both paradigms, hybrid approaches such as TimeXer (Wang et al., 2024) combine patch-level and variate-level representations to jointly model temporal dynamics and cross-variable dependencies. Despite the remarkable success of recent models, their effectiveness does not readily extend to IMTS. This limitation primarily stems from the implicit assumption of uniformly sampled observations, which leads to significant performance degradation in the presence of temporal irregularities (Chowdhury et al., 2023).

### 2.2  Irregular Multivariate Time Series Modeling

Existing methods for IMTS can be broadly categorized into irregularity-specific architectures and data-based approaches. The former approaches directly model the irregular nature of IMTS. RNN-based methods, such as GRU-D (Che et al., 2018), DATA-GRU (Tan et al., 2020), and P-LSTM (Neil et al., 2016), address missing values and irregular sampling through decay mechanisms, attention modules, or time-gated updates. Continuous-time models, including ODE-RNN, Latent-ODE (Rubanova et al., 2019), and ContiFormer (Chen et al., 2023b), employ neural ODEs to capture latent dynamics evolving continuously between observations. GNN-based methods capture complex temporal and relational dependencies by leveraging dynamic and time-adaptive graph structures, such as GraFITi (Yalavarthi et al., 2024) and tPatchGNN (Zhang et al., 2024). Furthermore, Hi-Patch (Luo et al., 2025) adopts a hierarchical multi-scale architecture, Raindrop (Zhang et al., 2022) incorporates sensor graphs, and HyperIMTS (Li et al., 2025) leverages hypergraph representations. In contrast, data-based approaches transform IMTS into regular temporal grids at either the raw-data or representation level. IP-Nets (Shukla & Marlin, 2019) perform semi-parametric interpolation at the data level, while mTAND (Shukla & Marlin, 2021) employs time-attention mechanisms to generate regular temporal latent representations. Despite their effectiveness, existing IMTS

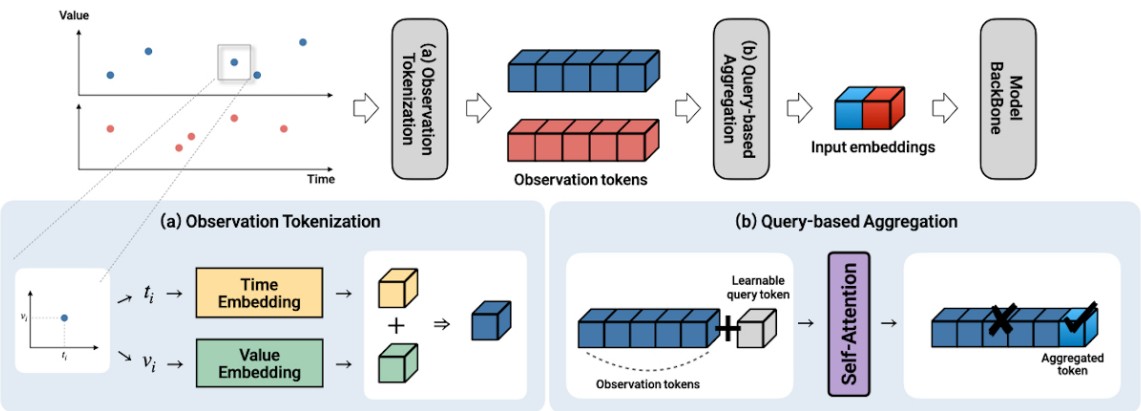

*Figure 2.* **Overall Framework of QuITE.** QuITE is a plug-and-play embedding module for IMTS that uses learnable query tokens to aggregate irregular observations through a single self-attention layer and produce structured observation-summary tokens. This example illustrates variable-level aggregation.

methods exhibit inherent trade-offs. Irregularity-specific architectures restrict the reuse of powerful and extensively validated MTS models. Conversely, data-based approaches may introduce data distortions that obscure the true nature of IMTS, ultimately limiting their performance (Luo et al., 2025; Chang et al., 2026).

## 3 Preliminaries

**Notation.** An IMTS instance is defined as

$$\mathcal{X} = \{(x_{n,i}, t_{n,i}, m_{n,i}) \mid n = 1, \ldots, N; \ i = 1, \ldots, L_n\}, \quad (1)$$

where $N$ denotes the number of variables and $L_n$ denotes the number of observations for variable $n$. Observations across different variables are not temporally aligned. For each variable $n$ and observation index $i$, $x_{n,i} \in \mathbb{R}$ denotes the observed value, $t_{n,i} \in \mathbb{R}^+$ denotes the corresponding continuous timestamp, and $m_{n,i} \in \{0, 1\}$ denotes a binary observation mask indicating whether the value is observed.

**Irregular Multivariate Time Series Classification.** Given an IMTS dataset $\{(\mathcal{X}_i, y_i)\}_{i=1}^K$, where $\mathcal{X}_i$ denotes the $i$-th IMTS instance and $y_i$ is its class label, the objective is to learn a classification model

$$C(\mathcal{X}_i) = \hat{y}_i, \quad (2)$$

where $C(\cdot)$ denotes the classification model and $\hat{y}_i$ denotes the predicted class label for $\mathcal{X}_i$.

**Irregular Multivariate Time Series Forecasting.** Given an IMTS instance $\mathcal{X}$, the objective of IMTS forecasting is to predict future observations beyond a forecasting start time $t_s$. Observations with timestamps earlier than $t_s$ are treated as historical inputs, while those occur-

ring at or after $t_s$ are treated as prediction targets. Formally, the historical and future observations are defined as $\mathcal{X}_{\text{hist}} = \{(x_{n,i}, t_{n,i}, m_{n,i}) \in \mathcal{X} \mid t_{n,i} < t_s\}$, and $\mathcal{X}_{\text{fut}} = \{(x_{n,i}, t_{n,i}, m_{n,i}) \in \mathcal{X} \mid t_{n,i} \geq t_s\}$, respectively. Given the historical IMTS sample $\mathcal{X}_{\text{hist}}$ and a forecast query set $Q$ specifying future timestamps, our goal is to learn a forecasting model

$$F(\mathcal{X}_{\text{hist}}, Q) = \hat{Y}, \quad (3)$$

where $\hat{Y}$ denotes the predicted values corresponding to the query timestamps in $Q$.

## 4 Query-Based Irregular Time Series Embedding

In this section, we present QuITE, a query-based input-embedding module designed for IMTS. We first introduce the observation tokenization method in Section 4.1, followed by variable-level and patch-level query-based aggregation modules in Section 4.2. An overview of the overall framework is illustrated in Figure 2.

### 4.1 Observation Tokenization

Given an IMTS instance $\mathcal{X}$, each entry is represented as a value–time–mask triplet $(x_{n,i}, t_{n,i}, m_{n,i})$.

**Time Embedding.** To encode continuous timestamps, we employ a harmonic time embedding function (Shukla & Marlin, 2021) $\phi : \mathbb{R}^+ \to \mathbb{R}^D$, defined as

$$\phi(t)[k] = \begin{cases} \omega_0 t + \alpha_0, & k = 0, \\ \sin(\omega_k t + \alpha_k), & k > 0, \end{cases} \quad (4)$$

where $\omega_k$ and $\alpha_k$ are learnable frequency and phase parameters.

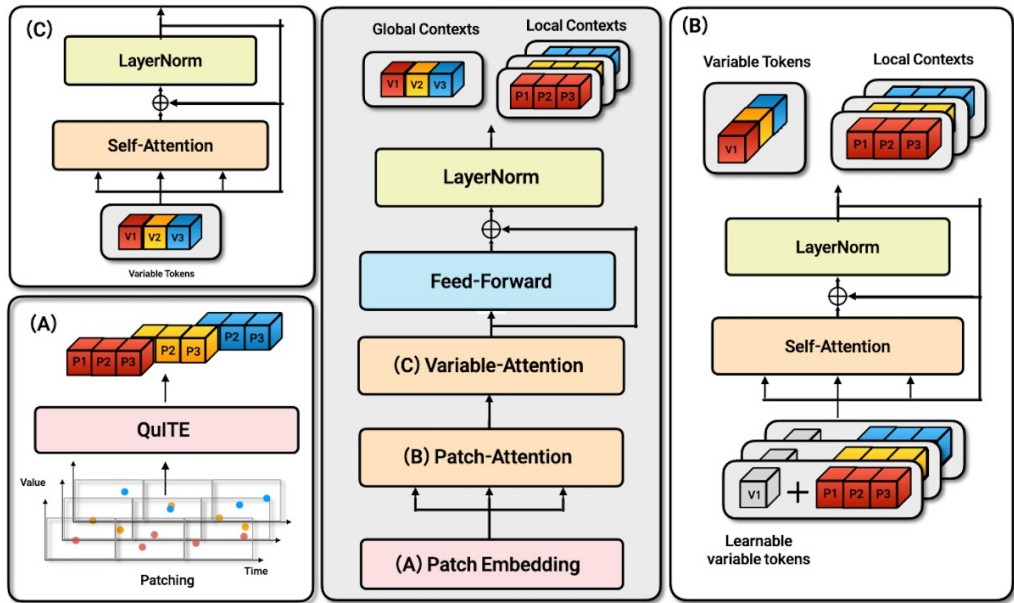

*Figure 3.* **Overall Architecture of QuITE$^{++}$.** A hierarchical encoder that models intra-variable patch-level temporal dependencies and inter-variable interactions via learnable query tokens.

**Value Embedding.** Each value is projected into the latent space via a linear encoder $f_{\text{val}} : \mathbb{R} \rightarrow \mathbb{R}^D$. The resulting observation token is defined as

$$\mathbf{z}_{n,i} = f_{\text{val}}(x_{n,i}) + \phi(t_{n,i}), \quad \mathbf{z}_{n,i} \in \mathbb{R}^D. \quad (5)$$

All tokens form an unordered observation-token set $\mathbf{Z} = \{\mathbf{z}_{n,i} \mid n = 1, \dots, N; \ i = 1, \dots, L_n\}$. The corresponding masks $\{m_{n,i}\}$ are used in the subsequent attention-based aggregation to exclude unobserved or padded entries.

### 4.2 Query-Based Aggregation

To transform the irregular observation-token set into structured representations, QuITE introduces a set of learnable query tokens. Each query token adaptively aggregates a subset of observation tokens via a single self-attention layer. We consider two aggregation strategies: variable-level and patch-level aggregation.

**Variable-level Aggregation.** For variable-level modeling, one query token is assigned to each variable:

$$\mathbf{q}_n \in \mathbb{R}^D, \quad n = 1, \dots, N. \quad (6)$$

Let $\mathbf{Z}_n \subset \mathbf{Z}$ denote the observation tokens associated with variable $n$. Aggregation is performed via masked self-attention:

$$\mathbf{H}_n = \text{SelfAttn}([\mathbf{q}_n; \mathbf{Z}_n], \mathbf{A}_n = [\mathbf{1} \mid \mathbf{m}_n]), \quad (7)$$

where $\mathbf{m}_n = \{m_{n,i}\}_{i=1}^{L_n}$ denotes the observation mask for variable $n$. The updated query token is used as the variable-

level embedding:

$$\mathbf{e}_n = \mathbf{H}_n[0]. \quad (8)$$

Stacking all variable-level embeddings yields

$$\mathbf{E}^{\text{var}} \in \mathbb{R}^{N \times D}. \quad (9)$$

**Patch-level Aggregation.** To capture local temporal patterns, QuITE can alternatively aggregate observations within temporal patches. For each patch–variable pair, a dedicated query token is introduced:

$$\mathbf{q}_{m,n} \in \mathbb{R}^D, \quad m = 1, \dots, M, \ n = 1, \dots, N. \quad (10)$$

Let $\mathbf{Z}_{m,n}$ and $\mathbf{m}_{m,n}$ denote the observation tokens and corresponding masks, respectively, for variable $n$ whose timestamps fall into patch $m$. Aggregation is performed independently for each patch–variable pair:

$$\mathbf{H}_{m,n} = \text{SelfAttn}([\mathbf{q}_{m,n}; \mathbf{Z}_{m,n}], \mathbf{A}_{m,n} = [\mathbf{1} \mid \mathbf{m}_{m,n}]). \quad (11)$$

The updated query token yields the patch-level representation:

$$\mathbf{e}_{m,n} = \mathbf{H}_{m,n}[0]. \quad (12)$$

The resulting structured embedding has shape

$$\mathbf{E}^{\text{patch}} \in \mathbb{R}^{M \times N \times D}, \quad (13)$$

where each token summarizes irregular observations within a local temporal region.

# 5 QuITE$^{++}$

Motivated by QuITE, QuITE$^{++}$ extends the learnable-query-token principle into a hierarchical forecasting architecture that captures both temporal dependencies and inter-variable interactions. An overview of the architecture is illustrated in Figure 3.

## 5.1 Hierarchical Encoder

For each variable $n \in \{1, \ldots, N\}$, we introduce a learnable *variable token* to summarize its global dynamics. The initial variable token is defined as

$$\mathbf{c}_n^{(0)} = \mathbf{v}_n \in \mathbb{R}^D, \tag{14}$$

where $\mathbf{v}_n$ denotes the variable identity embedding and $\mathbf{c}_n^{(0)}$ denotes the initial variable-level representation at layer 0.

Given the patch-level representations $\mathbf{E}^{\text{patch}}$ obtained from QuITE, we initialize $\mathbf{e}_{m,n}^{(0)}$ as the representation of patch $m$ for variable $n$. Let $\mathbf{e}_{m,n}^{(\ell)} \in \mathbb{R}^D$ denote the patch-level representation at layer $\ell$. The hierarchical encoder consists of $L$ stacked layers, each comprising two successive attention blocks.

**Patch-level Self-attention.** For each variable $n$, we first model temporal interactions across patches. The variable token is prepended to the patch sequence:

$$\mathbf{S}_n^{(\ell)} = \left[ \mathbf{c}_n^{(\ell-1)}; \mathbf{e}_{1,n}^{(\ell-1)}; \cdots; \mathbf{e}_{M,n}^{(\ell-1)} \right] \in \mathbb{R}^{(1+M) \times D}. \tag{15}$$

A patch-level self-attention layer is then applied:

$$\tilde{\mathbf{S}}_n^{(\ell)} = \text{SelfAttn}^{\text{patch}}{}_\ell \left( \mathbf{S}_n^{(\ell)} \right). \tag{16}$$

The updated variable token and patch representations are obtained as

$$\mathbf{c}_n^{(\ell)} = \tilde{\mathbf{S}}_n^{(\ell)}[0], \qquad \mathbf{e}_{m,n}^{(\ell)} = \tilde{\mathbf{S}}_n^{(\ell)}[m]. \tag{17}$$

This stage enables each variable to aggregate information across temporal patches, capturing local and mid-range temporal structures.

**Variable-level Self-attention.** After patch-level self-attention, the updated variable tokens $\{\mathbf{c}_n^{(\ell)}\}_{n=1}^N$ are treated as a variate-token sequence and processed by a variable-level self-attention layer:

$$\mathbf{C}^{(\ell)} = \text{SelfAttn}^{\text{var}}{}_\ell \left( \left[ \mathbf{c}_1^{(\ell)}, \ldots, \mathbf{c}_N^{(\ell)} \right] \right) \in \mathbb{R}^{N \times D}. \tag{18}$$

The variable-level representations are then updated as

$$\mathbf{c}_n^{(\ell)} = \mathbf{C}^{(\ell)}[n], \quad n = 1, \ldots, N. \tag{19}$$

After $L$ hierarchical layers, we obtain the final representations:

$$\mathbf{C} = \mathbf{C}^{(L)} \in \mathbb{R}^{N \times D}, \tag{20}$$

$$\mathbf{E} = \{\mathbf{e}_{m,n}^{(L)}\}_{m=1,n=1}^{M,N} \in \mathbb{R}^{M \times N \times D}. \tag{21}$$

## 5.2 Decoder

**Future Time Query Embedding.** Given future timestamps $\{\tau_j\}_{j=1}^{L_{\text{pred}}}$, we obtain their embeddings by applying the same time embedding function defined in Equation 4:

$$\mathbf{u}_j = \phi(\tau_j) \in \mathbb{R}^D, \qquad \mathbf{U} = [\mathbf{u}_1; \ldots; \mathbf{u}_{L_{\text{pred}}}] \in \mathbb{R}^{L_{\text{pred}} \times D}. \tag{22}$$

These embeddings serve as future-time queries in the cross-attention decoder. For each variable $n$, we extract two complementary contexts via cross-attention.

**Global Context.** The final variable-level representation $\mathbf{C}_n$ serves as a compact global summary for variable $n$:

$$\mathbf{G}_n = \text{CrossAttn}(\mathbf{U}, \mathbf{C}_n, \mathbf{C}_n) \in \mathbb{R}^{L_{\text{pred}} \times D}. \tag{23}$$

**Local Context.** To preserve fine-grained temporal patterns, the patch-level representations $\mathbf{E}_n = [\mathbf{e}_{1,n}; \ldots; \mathbf{e}_{M,n}] \in \mathbb{R}^{M \times D}$ are used as keys and values:

$$\mathbf{R}_n = \text{CrossAttn}(\mathbf{U}, \mathbf{E}_n, \mathbf{E}_n) \in \mathbb{R}^{L_{\text{pred}} \times D}. \tag{24}$$

**Final Prediction.** The global and local contexts are concatenated and passed through an MLP to produce the final prediction:

$$\hat{y}_{j,n} = f_{\text{out}} \left( [\mathbf{G}_n[j]; \mathbf{R}_n[j]] \right), \tag{25}$$

where $f_{\text{out}} : \mathbb{R}^{2D} \to \mathbb{R}$ denotes a three-layer feed-forward network. The final forecasting output is

$$\hat{\mathbf{Y}} = \{\hat{y}_{j,n}\}_{j=1,n=1}^{L_{\text{pred}},N} \in \mathbb{R}^{L_{\text{pred}} \times N}. \tag{26}$$

# 6 Experiments

## 6.1 Experimental Setups

**Dataset.** We evaluate our method on four forecasting benchmarks and three classification benchmarks. For forecasting, we use Human Activity (Kaluža et al., 2010), USHCN (Menne et al., 2015), PhysioNet (Silva et al., 2012), and MIMIC-III (Johnson et al., 2016), covering biomechanics, climate, and healthcare domains. For classification, we use healthcare and human activity datasets: P19 (Reyna et al., 2020), P12 (Goldberger et al., 2000), and PAM (Reiss & Stricker, 2012). More details are provided in Appendix A.

| Model | | Patch | | | | | | | | | | | | | | | | | |
|---|---|---|---|---|---|---|---|---|---|---|---|---|---|---|---|---|---|---|---|
| | | PatchTST | | + QuITE | | Imp. | | PatchMixer | | + QuITE | | Imp. | | TMix | | + QuITE | | Imp. | |
| Dataset | Horizon | MSE | MAE | MSE | MAE | MSE | MAE | MSE | MAE | MSE | MAE | MSE | MAE | MSE | MAE | MSE | MAE | MSE | MAE |
| Human Activity (ms) | 3000 → 1000 | 3.10 | 3.44 | 2.76 | 3.14 | +10.97% | +8.72% | 2.84 | 3.21 | 2.78 | 3.13 | +2.11% | +2.49% | 2.93 | 3.28 | 2.77 | 3.15 | +5.59% | +3.99% |
| | 2000 → 2000 | 4.02 | 3.99 | 3.62 | 3.74 | +9.95% | +6.27% | 3.76 | 3.84 | 3.67 | 3.75 | +2.39% | +2.34% | 3.87 | 3.66 | 3.74 | 3.74 | +5.47% | +4.40% |
| | 1000 → 3000 | 5.06 | 4.52 | 4.69 | 4.29 | +7.31% | +5.09% | 4.80 | 4.40 | 4.71 | 4.31 | +1.88% | +2.05% | 4.97 | 4.48 | 4.75 | 4.38 | +4.54% | +2.26% |
| USHCN (m) | 24 → 1 | 5.35 | 3.31 | 5.07 | 3.01 | +5.21% | +9.05% | 5.22 | 3.14 | 5.11 | 3.06 | +2.11% | +2.55% | 5.41 | 3.27 | 5.18 | 3.10 | +4.13% | +5.17% |
| | 24 → 6 | 5.30 | 3.32 | 5.06 | 3.17 | +4.49% | +4.64% | 5.31 | 3.13 | 5.02 | 3.04 | +5.46% | +2.88% | 7.31 | 4.49 | 5.52 | 3.39 | +24.52% | +24.41% |
| | 24 → 12 | 5.11 | 3.09 | 5.04 | 3.00 | +1.46% | +2.97% | 5.22 | 3.12 | 5.00 | 3.07 | +4.21% | +1.60% | 7.53 | 4.56 | 5.03 | 3.05 | +33.18% | +33.12% |
| PhysioNet (h) | 12 → 36 | 20.03 | 8.16 | 17.47 | 7.15 | +12.78% | +12.38% | 18.62 | 7.64 | 17.52 | 7.22 | +5.91% | +5.50% | 18.57 | 7.65 | 17.48 | 7.21 | +5.86% | +5.81% |
| | 24 → 24 | 15.28 | 7.02 | 10.62 | 5.17 | +30.47% | +26.37% | 12.28 | 5.88 | 11.88 | 5.62 | +3.26% | +4.42% | 12.38 | 5.81 | 10.72 | 5.22 | +13.39% | +10.17% |
| | 36 → 12 | 12.97 | 6.07 | 8.87 | 4.43 | +31.58% | +27.04% | 10.33 | 5.03 | 9.06 | 4.55 | +12.29% | +9.54% | 10.50 | 5.13 | 8.96 | 4.50 | +14.74% | +12.31% |
| MIMIC-III (h) | 12 → 36 | 5.67 | 17.62 | 5.37 | 17.01 | +5.29% | +3.44% | 5.53 | 17.23 | 5.37 | 16.96 | +2.89% | +1.57% | 5.57 | 17.30 | 5.41 | 16.94 | +2.81% | +2.10% |
| | 24 → 24 | 4.33 | 13.28 | 3.90 | 12.35 | +9.87% | +7.04% | 4.10 | 12.62 | 3.94 | 12.34 | +3.90% | +2.22% | 4.14 | 12.77 | 3.98 | 12.32 | +4.04% | +3.52% |
| | 36 → 12 | 4.86 | 14.12 | 3.82 | 11.84 | +21.36% | +16.16% | 4.11 | 12.10 | 3.87 | 11.51 | +5.84% | +4.88% | 4.04 | 12.05 | 3.87 | 11.63 | +4.27% | +3.51% |
| Average | | 7.59 | 7.33 | 6.36 | 6.52 | +16.20% | +11.00% | 6.84 | 6.78 | 6.49 | 6.55 | +5.10% | +3.40% | 7.27 | 7.06 | 6.44 | 6.55 | +11.40% | +7.20% |

| Model | | Variate | | | | | | | | | | | | Hybrid | | | | | |
|---|---|---|---|---|---|---|---|---|---|---|---|---|---|---|---|---|---|---|---|
| | | iTransformer | | + QuITE | | Imp. | | S-Mamba | | + QuITE | | Imp. | | TimeXer | | + QuITE | | Imp. | |
| Dataset | Horizon | MSE | MAE | MSE | MAE | MSE | MAE | MSE | MAE | MSE | MAE | MSE | MAE | MSE | MAE | MSE | MAE | MSE | MAE |
| Human Activity (ms) | 3000 → 1000 | 3.77 | 4.13 | 2.58 | 3.12 | +31.48% | +24.50% | 3.56 | 4.05 | 2.72 | 3.24 | +23.60% | +20.00% | 2.99 | 3.41 | 2.53 | 3.04 | +15.52% | +10.86% |
| | 2000 → 2000 | 4.88 | 4.80 | 3.25 | 3.65 | +33.46% | +23.94% | 4.68 | 4.77 | 3.37 | 3.72 | +27.99% | +22.01% | 3.87 | 3.96 | 3.19 | 3.57 | +17.47% | +9.79% |
| | 1000 → 3000 | 5.71 | 5.20 | 4.10 | 4.18 | +28.15% | +19.64% | 5.37 | 5.10 | 4.20 | 4.22 | +21.79% | +17.25% | 4.77 | 4.50 | 4.04 | 4.03 | +15.30% | +10.42% |
| USHCN (m) | 24 → 1 | 5.91 | 3.63 | 5.06 | 3.06 | +14.37% | +15.67% | 5.73 | 3.57 | 5.04 | 3.06 | +12.04% | +14.29% | 5.36 | 3.24 | 4.97 | 2.97 | +7.30% | +8.19% |
| | 24 → 6 | 5.51 | 3.22 | 4.86 | 2.96 | +11.77% | +7.96% | 5.50 | 3.25 | 4.93 | 3.04 | +10.36% | +6.46% | 5.21 | 3.17 | 4.98 | 3.05 | +4.46% | +3.86% |
| | 24 → 12 | 5.46 | 3.21 | 4.94 | 3.01 | +9.50% | +6.17% | 5.40 | 3.20 | 4.93 | 3.02 | +8.70% | +5.63% | 5.23 | 3.07 | 4.93 | 2.97 | +5.80% | +3.37% |
| PhysioNet (h) | 12 → 36 | 21.14 | 8.74 | 6.32 | 4.15 | +70.11% | +52.49% | 8.23 | 4.71 | 6.26 | 4.11 | +23.94% | +12.55% | 6.91 | 4.43 | 6.18 | 4.08 | +10.56% | +7.99% |
| | 24 → 24 | 16.48 | 7.94 | 4.99 | 3.65 | +69.72% | +54.06% | 6.93 | 4.40 | 5.11 | 3.67 | +26.26% | +16.59% | 5.79 | 4.14 | 4.91 | 3.64 | +15.20% | +12.10% |
| | 36 → 12 | 14.09 | 6.68 | 4.33 | 3.34 | +69.27% | +50.03% | 6.26 | 4.25 | 4.11 | 3.27 | +34.35% | +23.06% | 4.94 | 3.72 | 4.06 | 3.27 | +17.85% | +12.05% |
| MIMIC-III (h) | 12 → 36 | 6.00 | 19.06 | 1.83 | 7.64 | +69.50% | +59.92% | 2.83 | 10.89 | 1.82 | 7.56 | +35.69% | +30.58% | 2.01 | 8.44 | 1.84 | 7.67 | +8.56% | +9.12% |
| | 24 → 24 | 5.50 | 17.20 | 1.67 | 6.93 | +69.64% | +59.70% | 2.38 | 9.38 | 1.64 | 6.90 | +31.09% | +26.44% | 1.92 | 7.98 | 1.68 | 7.12 | +12.42% | +10.77% |
| | 36 → 12 | 6.05 | 17.84 | 1.56 | 6.78 | +74.19% | +62.00% | 3.92 | 11.74 | 1.52 | 6.64 | +61.22% | +43.44% | 1.98 | 8.64 | 1.55 | 6.73 | +21.67% | +22.07% |
| Average | | 8.37 | 8.47 | 3.79 | 4.37 | +54.70% | +48.40% | 5.07 | 5.78 | 3.80 | 4.37 | +25.00% | +24.30% | 4.25 | 4.89 | 3.74 | 4.35 | +12.00% | +11.20% |

*Table 2.* **Effectiveness of QuITE in Forecasting.** "Imp." represents the relative percentage improvement over the original backbone model after applying QuITE.

| Dataset | Metric | Patch | | | | | |
|---|---|---|---|---|---|---|---|
| | | PatchTST | + QuITE | Imp. | PatchMixer | + QuITE | Imp. |
| P12 | AUROC | 83.6 | 84.5 | 1.1% | 78.2 | 83.9 | 7.3% |
| | AUPRC | 47.0 | 54.5 | 16.0% | 39.2 | 45.2 | 15.3% |
| P19 | AUROC | 82.4 | 85.0 | 3.2% | 75.0 | 83.8 | 11.7% |
| | AUPRC | 40.3 | 54.5 | 35.2% | 26.4 | 55.8 | 111.4% |
| PAM | Accuracy | 84.7 | 88.1 | 4.0% | 73.5 | 82.2 | 11.8% |
| | Precision | 86.4 | 89.6 | 3.7% | 77.6 | 86.4 | 11.3% |
| | Recall | 86.5 | 89.3 | 3.2% | 75.6 | 82.7 | 9.4% |
| | F1 score | 86.3 | 89.3 | 3.5% | 75.7 | 83.7 | 10.6% |
| Average | | 74.7 | 79.4 | 6.3% | 65.2 | 75.5 | 15.8% |

| Dataset | Metric | Patch | | | Variate | | |
|---|---|---|---|---|---|---|---|
| | | TMix | + QuITE | Imp. | iTransformer | + QuITE | Imp. |
| P12 | AUROC | 82.8 | 85.2 | 2.9% | 82.6 | 85.3 | 3.3% |
| | AUPRC | 45.5 | 51.9 | 14.1% | 45.3 | 47.6 | 5.1% |
| P19 | AUROC | 63.0 | 81.2 | 28.9% | 82.5 | 86.5 | 4.8% |
| | AUPRC | 10.1 | 38.0 | 276.2% | 39.2 | 51.7 | 31.9% |
| PAM | Accuracy | 80.0 | 84.3 | 5.4% | 87.6 | 90.7 | 3.5% |
| | Precision | 82.0 | 86.1 | 5.0% | 89.3 | 91.8 | 2.8% |
| | Recall | 81.3 | 86.1 | 5.9% | 89.3 | 91.4 | 2.4% |
| | F1 score | 81.4 | 85.9 | 5.5% | 89.2 | 91.5 | 2.6% |
| Average | | 65.8 | 74.9 | 13.8% | 75.6 | 79.6 | 5.3% |

| Dataset | Metric | Variate | | | Hybrid | | |
|---|---|---|---|---|---|---|---|
| | | S-Mamba | + QuITE | Imp. | TimeXer | + QuITE | Imp. |
| P12 | AUROC | 82.3 | 84.1 | 2.2% | 83.5 | 86.1 | 3.1% |
| | AUPRC | 43.6 | 49.5 | 13.5% | 48.0 | 49.5 | 3.1% |
| P19 | AUROC | 80.2 | 85.4 | 6.5% | 83.2 | 85.3 | 2.5% |
| | AUPRC | 37.0 | 51.7 | 39.7% | 41.9 | 54.9 | 31.0% |
| PAM | Accuracy | 85.5 | 87.4 | 2.2% | 85.6 | 89.4 | 4.4% |
| | Precision | 86.1 | 89.0 | 3.4% | 87.0 | 90.2 | 3.7% |
| | Recall | 85.7 | 88.7 | 3.5% | 87.6 | 90.5 | 3.3% |
| | F1 score | 85.9 | 88.7 | 3.3% | 87.0 | 90.3 | 3.8% |
| Average | | 73.3 | 78.1 | 6.5% | 75.5 | 79.5 | 5.3% |

*Table 3.* **Effectiveness of QuITE in Classification.** "Imp." represents the relative percentage improvement over the original backbone model after applying QuITE.

**Baselines.** We compare our method against 17 representative baselines to comprehensively evaluate its performance on IMTS. These include state-of-the-art and widely used methods from two categories: (i) MTS forecasting models, including PatchTST (Nie et al., 2022), PatchMixer (Gong et al., 2023), TMix (Chen et al., 2023a), iTransformer (Liu et al., 2023), S-Mamba (Wang et al., 2025), and TimeXer (Wang et al., 2024); and (ii) IMTS models, including Warpformer (Zhang et al., 2023), GRU-D (Che et al., 2018), Hi-Patch (Luo et al., 2025), mTAND (Shukla & Marlin, 2021), Raindrop (Zhang et al., 2022), tPatchGNN (Zhang et al., 2024), CRU (Schirmer et al., 2022), NeuralFlow (Biloš et al., 2021), Latent ODE (Rubanova et al., 2019), HyperIMTS (Li et al., 2025), and GraFITi (Yalavarthi et al., 2024). More detailed descriptions of all baselines are provided in Appendix B.

**Implementation Details.** All experiments are repeated over five random seeds ($\{1, \ldots, 5\}$), and early stopping is applied when the validation loss does not improve for 50 epochs. We use cross-entropy loss for classification and mean squared error (MSE) loss for forecasting. For QuITE-equipped MTS backbones, we use four attention heads and a hidden dimension of 64. The number of learnable query tokens is determined by the target structure required by each backbone, rather than being tuned as a hyperparameter. For QuITE$^{++}$, we tune the hidden dimension, number of layers, and number of attention heads via grid search over $\{32, 64\}$,

# QuITE: Query-Based Irregular Time Series Embedding

| Dataset | Human Activity (ms) | | | | | | USHCN (m) | | | | | |
|---|---|---|---|---|---|---|---|---|---|---|---|---|
| Horizon | 3000 → 1000 | | 2000 → 2000 | | 1000 → 3000 | | 24 → 1 | | 24 → 6 | | 24 → 12 | |
| Metric | MSE | MAE | MSE | MAE | MSE | MAE | MSE | MAE | MSE | MAE | MSE | MAE |
| Warpformer | 2.61 | 3.12 | 3.60 | 3.81 | 4.26 | 4.26 | 5.09 | 3.10 | 5.12 | 3.13 | 5.10 | 3.13 |
| Raindrop | 4.42 | 4.65 | 5.57 | 5.15 | 5.75 | 5.37 | 5.64 | 3.29 | 7.01 | 4.24 | 7.61 | 4.61 |
| GRU-D | 3.94 | 4.37 | 5.93 | 5.66 | 6.14 | 5.75 | 5.17 | 3.21 | 5.29 | 3.34 | 5.36 | 3.25 |
| tPatchGNN | 2.79 | 3.24 | 3.71 | 3.89 | 4.56 | 4.32 | 5.00 | 3.07 | 5.23 | 3.24 | 6.23 | 3.83 |
| GraFITi | 3.03 | 3.45 | 4.59 | 4.45 | 4.91 | 4.62 | 5.07 | 2.97 | 5.12 | 3.09 | 5.01 | 3.14 |
| CRU | 3.03 | 3.60 | 4.12 | 4.43 | 4.85 | 4.86 | 5.15 | 3.18 | 6.77 | 4.11 | 6.64 | 4.08 |
| mTAND | 3.14 | 3.71 | 4.38 | 4.59 | 5.29 | 5.12 | 5.03 | 3.00 | 5.16 | 3.10 | 5.07 | 3.09 |
| NeuralFlow | 4.29 | 4.61 | 5.47 | 5.35 | 6.01 | 5.66 | 5.41 | 3.35 | 5.52 | 3.46 | 5.48 | 3.56 |
| Latent-ODE | 3.32 | 3.91 | 5.04 | 5.11 | 5.48 | 5.33 | 5.16 | 3.21 | 5.18 | 3.36 | 5.23 | 3.35 |
| HyperIMTS | 2.49 | 3.02 | 3.15 | 3.58 | 4.00 | 4.13 | 4.96 | 3.00 | 4.99 | 3.10 | 4.97 | 3.08 |
| Hi-Patch | 2.56 | 3.12 | 3.26 | 3.67 | 4.20 | 4.22 | 5.00 | 3.03 | 5.13 | 3.05 | 5.04 | 3.03 |
| PatchTST + QuITE | 2.76 | 3.14 | 3.62 | 3.74 | 4.69 | 4.29 | 5.07 | 3.01 | 5.06 | 3.17 | 5.04 | 3.00 |
| PatchMixer + QuITE | 2.78 | 3.13 | 3.67 | 3.75 | 4.71 | 4.31 | 5.11 | 3.06 | 5.02 | 3.04 | 5.00 | 3.07 |
| TMix + QuITE | 2.77 | 3.15 | 3.66 | 3.74 | 4.75 | 4.38 | 5.18 | 3.10 | 5.52 | 3.39 | 5.03 | 3.05 |
| iTransformer + QuITE | 2.58 | 3.12 | 3.25 | 3.65 | 4.10 | 4.18 | 5.06 | 3.06 | 4.86 | 2.96 | 4.94 | 3.01 |
| S-Mamba + QuITE | 2.72 | 3.24 | 3.37 | 3.72 | 4.20 | 4.22 | 5.04 | 3.06 | 4.93 | 3.04 | 4.93 | 3.02 |
| TimeXer + QuITE | 2.53 | 3.04 | 3.19 | 3.57 | 4.04 | 4.03 | 4.97 | 2.97 | 4.98 | 3.05 | 4.93 | 2.97 |
| QuITE$^{++}$ | 2.46 | 2.92 | 3.11 | 3.49 | 3.96 | 4.04 | 4.84 | 2.92 | 4.81 | 2.94 | 4.81 | 2.93 |

| Dataset | PhysioNet (h) | | | | | | MIMIC-III (h) | | | | | |
|---|---|---|---|---|---|---|---|---|---|---|---|---|
| Horizon | 12 → 36 | | 24 → 24 | | 36 → 12 | | 12 → 36 | | 24 → 24 | | 36 → 12 | |
| Metric | MSE | MAE | MSE | MAE | MSE | MAE | MSE | MAE | MSE | MAE | MSE | MAE |
| Warpformer | 6.51 | 4.24 | 5.04 | 3.72 | 4.17 | 3.38 | 2.32 | 8.14 | 1.76 | 7.27 | 1.45 | 6.74 |
| Raindrop | 10.24 | 5.83 | 10.63 | 6.02 | 10.67 | 5.87 | 2.36 | 8.63 | 2.31 | 8.61 | 2.21 | 9.17 |
| GRU-D | 7.80 | 5.13 | 5.76 | 4.53 | 6.85 | 4.88 | 2.39 | 8.43 | 2.35 | 8.34 | 2.03 | 8.14 |
| tPatchGNN | 6.45 | 4.24 | 5.06 | 3.75 | 4.22 | 3.38 | 2.35 | 8.23 | 1.97 | 7.76 | 1.44 | 6.78 |
| GraFITi | 6.30 | 4.38 | 5.11 | 3.96 | 4.58 | 3.65 | 2.22 | 8.13 | 1.76 | 7.28 | 1.61 | 7.16 |
| CRU | 7.66 | 4.97 | 6.43 | 4.51 | 6.74 | 4.82 | 2.34 | 8.32 | 2.23 | 7.99 | 2.00 | 8.16 |
| mTAND | 7.46 | 4.85 | 6.18 | 4.44 | 5.61 | 4.15 | 2.29 | 8.38 | 2.15 | 8.00 | 2.01 | 8.13 |
| NeuralFlow | 7.98 | 5.08 | 7.68 | 4.84 | 8.87 | 5.43 | 2.26 | 8.29 | 2.34 | 8.09 | 1.97 | 8.39 |
| Latent-ODE | 7.28 | 4.83 | 6.85 | 4.77 | 6.99 | 4.74 | 2.38 | 8.35 | 2.11 | 7.76 | 1.90 | 7.92 |
| HyperIMTS | 6.11 | 4.23 | 4.65 | 3.56 | 3.99 | 3.21 | 1.85 | 7.71 | 1.68 | 6.92 | 1.52 | 6.68 |
| Hi-Patch | 6.39 | 4.10 | 5.07 | 3.63 | 4.27 | 3.30 | 1.88 | 7.95 | 1.70 | 7.18 | 1.56 | 6.74 |
| PatchTST + QuITE | 17.47 | 7.15 | 10.62 | 5.17 | 8.87 | 4.43 | 5.37 | 17.01 | 3.90 | 12.35 | 3.82 | 11.84 |
| PatchMixer + QuITE | 17.52 | 7.22 | 11.88 | 5.62 | 9.06 | 4.55 | 5.37 | 16.96 | 3.94 | 12.34 | 3.87 | 11.51 |
| TMix + QuITE | 17.48 | 7.21 | 10.72 | 5.22 | 8.96 | 4.50 | 5.41 | 16.94 | 3.98 | 12.32 | 3.87 | 11.63 |
| iTransformer + QuITE | 6.32 | 4.15 | 4.99 | 3.65 | 4.33 | 3.34 | 1.83 | 7.64 | 1.67 | 6.93 | 1.56 | 6.78 |
| S-Mamba + QuITE | 6.26 | 4.11 | 5.11 | 3.67 | 4.11 | 3.27 | 1.82 | 7.56 | 1.64 | 6.90 | 1.52 | 6.64 |
| TimeXer + QuITE | 6.18 | 4.08 | 4.91 | 3.64 | 4.06 | 3.27 | 1.84 | 7.67 | 1.68 | 7.12 | 1.55 | 6.73 |
| QuITE$^{++}$ | 6.08 | 3.99 | 4.99 | 3.62 | 3.81 | 3.18 | 1.80 | 7.54 | 1.63 | 6.83 | 1.48 | 6.56 |

*Table 4.* **Forecasting performance comparison against baselines.** QuITE$^{++}$ achieves the strongest overall performance, while QuITE-equipped MTS backbones show competitive performance against baselines. The best and second-best results are highlighted in red and blue, respectively.

$\{1, 2, 3\}$, and $\{1, 2, 4, 8\}$, respectively. All experiments are conducted on an NVIDIA RTX A6000 GPU. More details are provided in Appendix C.

## 6.2 Main Results

Forecasting performance is evaluated using mean squared error (MSE) and mean absolute error (MAE) under three observation lengths and forecast horizons for each dataset. To ensure a fair comparison, we use the same cross-attention decoder as in QuITE$^{++}$. Unlike an MLP decoder, which may require additional pooling or flattening for patch-based models, the cross-attention decoder allows both patch- and variable-based models to condition predictions directly on future time queries. For classification, we report AUROC and AUPRC on the imbalanced P12 and P19 datasets, and accuracy, precision, recall, and F1 score on the relatively balanced PAM dataset. The decoder flattens the outputs and feeds them into a three-layer MLP. Detailed averages and standard deviations are provided in Appendix D.

**Effectiveness of QuITE.** We evaluate the effectiveness of QuITE on representative MTS backbones spanning different tokenization strategies and backbone architectures: patch-token models, including PatchTST (Transformer), PatchMixer (CNN), and TMix (MLP); variate-token models, including iTransformer (Transformer) and S-Mamba (Mamba); and the hybrid model TimeXer (Transformer). We replace only the input embedding module with QuITE, while leaving the rest of the architecture unchanged. Detailed experimental settings are provided in Appendix B.

For forecasting, Table 2 shows that QuITE consistently improves performance across all tokenization strategies and backbone architectures, with average gains ranging from 5.1% to 54.7%. Variate-token models benefit more from QuITE, as they represent the entire sequence at the variable level and are therefore more sensitive to irregular sampling. For classification, Table 3 shows that QuITE also consistently improves the corresponding backbones, with mean relative gains ranging from approximately 5.3% to 15.8%

| Model | | | PatchTST | | | | iTransformer | | | | QuITE++ | | | |
|---|---|---|---|---|---|---|---|---|---|---|---|---|---|---|
| Method | Metric | | Activity | USHCN | PhysioNet | MIMIC-III | Activity | USHCN | PhysioNet | MIMIC-III | Activity | USHCN | PhysioNet | MIMIC-III |
| Add | MSE | | 4.00 | 5.23 | 13.79 | 4.71 | 4.98 | 6.26 | 18.26 | 6.34 | 3.44 | 5.05 | 5.34 | 1.71 |
| | MAE | | 4.03 | 3.17 | 6.54 | 14.74 | 4.84 | 3.80 | 8.01 | 19.73 | 3.76 | 3.04 | 3.86 | 7.31 |
| Concat | MSE | | 3.90 | 5.21 | 12.97 | 4.52 | 5.77 | 6.10 | 18.27 | 6.48 | 3.35 | 4.99 | 5.43 | 1.75 |
| | MAE | | 3.91 | 3.18 | 5.96 | 14.13 | 5.21 | 3.67 | 8.01 | 20.28 | 3.71 | 3.02 | 3.90 | 7.28 |
| mTAND | MSE | | 3.74 | 5.21 | 13.38 | 4.39 | 3.50 | 5.23 | 13.11 | 4.34 | 3.34 | 5.01 | 4.96 | 1.71 |
| | MAE | | 3.76 | 3.20 | 6.03 | 13.80 | 3.65 | 3.15 | 5.96 | 13.72 | 3.64 | 3.07 | 3.60 | 7.17 |
| Mean Pooling | MSE | | 3.75 | 5.14 | 12.84 | 4.43 | 3.59 | 5.08 | 12.15 | 4.33 | 3.31 | 4.94 | 5.26 | 1.69 |
| | MAE | | 3.77 | 3.19 | 5.95 | 13.84 | 3.72 | 3.11 | 5.70 | 13.61 | 3.64 | 3.01 | 3.89 | 7.23 |
| QuITE | MSE | | 3.69 | 5.06 | 12.32 | 4.36 | 3.31 | 4.95 | 5.21 | 1.69 | 3.18 | 4.82 | 4.96 | 1.64 |
| | MAE | | 3.72 | 3.06 | 5.58 | 13.73 | 3.65 | 3.01 | 3.71 | 7.12 | 3.48 | 2.93 | 3.60 | 6.98 |

*Table 5.* **Horizon-Average forecasting performance of different embedding methods.** The best and second-best results are highlighted in **red** and blue, respectively.

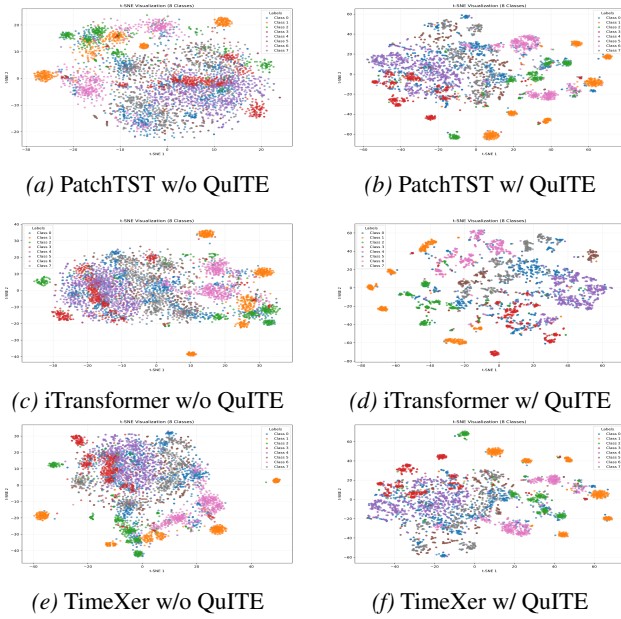

*(a)* PatchTST w/o QuITE    *(b)* PatchTST w/ QuITE

*(c)* iTransformer w/o QuITE    *(d)* iTransformer w/ QuITE

*(e)* TimeXer w/o QuITE    *(f)* TimeXer w/ QuITE

*Figure 4.* t-SNE Visualization of Embedding Representations on PAM.

across backbone families. These results indicate that modeling irregularity at the input stage is crucial for adapting MTS backbones to IMTS. They further demonstrate that QuITE serves as an effective backbone-agnostic embedding module for IMTS.

**Forecasting.** Table 4 compares the forecasting performance of 11 IMTS-specific models, six QuITE-equipped MTS models, and our proposed QuITE++. QuITE++ achieves the best performance in 20 out of 24 settings, demonstrating its strong and robust performance across diverse IMTS benchmarks. Although some IMTS-specific models, such as HyperIMTS, show competitive performance on certain datasets or horizons, QuITE++ consistently achieves strong results across evaluation scenarios. MTS backbones equipped with QuITE also achieve competitive performance, ranking first in one setting and second in

12 settings. These results indicate that replacing the input embedding module with QuITE substantially improves the ability of MTS models to handle IMTS without requiring architectural modification or artificial value generation.

On PhysioNet and MIMIC-III, the relatively weaker performance of QuITE-equipped patch-based models appears to stem from their limited inter-variable modeling capability rather than from QuITE itself. Patch-based models largely rely on variable-independent modeling, which can be restrictive for clinical IMTS datasets where inter-variable interactions are important (Zhang et al., 2022). This is further supported by TimeXer, which shares a patch-based foundation with PatchTST but achieves substantially stronger performance by explicitly modeling inter-variable interactions. Thus, the final gains from QuITE depend on how effectively the downstream backbone can exploit the adapted representations.

### 6.3 Analysis

**Comparison of Different Embedding Methods.** Table 5 compares different embedding strategies for adapting MTS models to IMTS, evaluated on PatchTST, iTransformer, and QuITE++. Specifically, Add and Concat incorporate temporal information by adding time embeddings to value embeddings and by concatenating values and timestamps, respectively, before passing them through conventional embedding layers. mTAND transforms IMTS into regular temporal grids in the latent space. Meanwhile, Mean Pooling first applies self-attention without learnable query tokens to obtain observation-level outputs (e.g., $[B, N, L, D]$ or $[B, N, M, L, D]$), and then applies mean pooling to construct the structured inputs expected by MTS backbones (e.g., $[B, N, D]$ or $[B, N, M, D]$).

As shown in Table 5, even simple input-level adaptations such as Add and Concat improve performance, validating the importance of modeling irregularity at the input stage. Building on this, QuITE achieves the best average forecasting performance across all datasets and backbones. Compared to Mean Pooling, which may dilute informative obser-

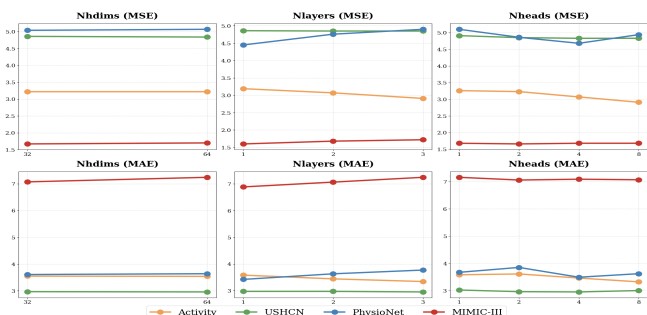

Figure 5. Hyperparameter sensitivity across hidden dimensions, numbers of layers and attention heads.

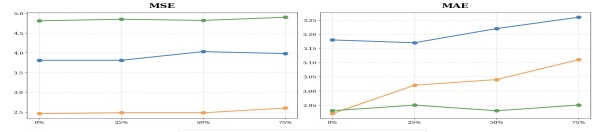

Figure 6. Robustness under additional random observation removal ratios.

| Dataset | Metric | Xavier | Uniform | Zero | Random |
|---|---|---|---|---|---|
| Human Activity | MSE | 2.46 | 2.45 | 2.46 | 2.46 |
| (3000ms → 1000ms) | MAE | 3.00 | 2.99 | 3.01 | 2.92 |
| USHCN | MSE | 4.83 | 4.86 | 4.87 | 4.81 |
| (24m → 12m) | MAE | 2.99 | 2.95 | 2.98 | 2.93 |

Table 6. Robustness to Query Initialization.

vations through uniform aggregation, QuITE summarizes irregular inputs into backbone-compatible structured embeddings, thereby bypassing the need for lossy pooling. Furthermore, unlike mTAND, QuITE operates directly on raw irregular observations without generating artificial values. Overall, these results demonstrate that QuITE is an effective embedding mechanism for adapting MTS models to IMTS. Detailed averages and standard deviations are provided in Appendix D.

**Quality of Embedding Representation.** To assess the quality of the learned embeddings, we visualize the model embedding vectors using t-SNE (Van der Maaten & Hinton, 2008) on the PAM dataset, which contains eight activity categories. We compare the embeddings before and after applying QuITE. As shown in Figure 4, QuITE produces more compact and clearly separated clusters across patch-level, variable-level, and hybrid embeddings. This indicates that QuITE improves the consistency and discriminative power of the learned embeddings regardless of the underlying tokenization strategy. Additional visualizations for other models are provided in Appendix F.1.

**Hyperparameter Sensitivity.** We analyze the sensitivity of QuITE$^{++}$ to the hidden dimension, number of layers, and number of attention heads using horizon-averaged MSE and MAE across four datasets. As shown in Figure 5, QuITE$^{++}$ remains generally robust across different hyperparameter choices.

**Robustness to Observation Sparsity.** We examine the practical limits of QuITE by randomly removing observations at different rates and evaluating QuITE$^{++}$ on forecasting tasks. We use representative settings from Human Activity, USHCN, and PhysioNet, corresponding to 3000ms → 1000ms, 24m → 1m, and 36h → 12h, respectively. As shown in Figure 6, performance either degrades gradually as the removal ratio increases or remains relatively stable up to about 50% additional removal, demonstrating robustness to sparse IMTS. At a 75% removal ratio, however, the

performance drop becomes substantially larger, suggesting a practical sparsity limit beyond which reliable forecasting becomes difficult.

**Robustness to Query Initialization.** By default, learnable query tokens are initialized using standard random initialization. We further evaluate Xavier, uniform, and zero initialization to assess sensitivity to query initialization. As shown in Table 6, QuITE remains robust across initialization schemes, with only minor performance differences.

**Computational Complexity Analysis.** We analyze the computational complexity of QuITE-equipped models in terms of parameter count, FLOPs, and training/inference time per epoch. In controlled comparisons, QuITE-equipped models use fewer parameters than their corresponding backbones, demonstrating that the gains are not simply due to increased model capacity. While QuITE generally reduces FLOPs alongside parameter counts, actual runtime may vary depending on backbone and implementation factors. Nevertheless, the improved performance on IMTS indicates a favorable accuracy–complexity trade-off. Furthermore, QuITE$^{++}$ remains competitive with strong IMTS-specific baselines in terms of performance, model size, and runtime. Detailed results are provided in Appendix E.

**Forecasting Visualization.** Additional forecasting visualizations with and without QuITE are provided in Appendix F.2.

## 7 Conclusion

This work presents QuITE, a plug-and-play input-embedding module that enables existing MTS models to process IMTS directly without architectural changes or artificial value generation. Extensive experiments demonstrate consistent performance improvements, highlighting the effectiveness of input-level adaptation for IMTS modeling. We hope this work encourages further exploration of flexible embedding-based approaches for IMTS modeling.

# Acknowledgements

We thank Daheen Kim and Seunghan Lee for their insightful discussions and helpful feedback on an earlier draft of this manuscript. We are also grateful to Prof. Changhee Lee, Dr. Jaeho Kim, Seongjun Lee, and Seokhyun Lee of Korea University, as well as Prof. Kyungwoo Song of Yonsei University, for their valuable suggestions and feedback. Finally, we highly appreciate the anonymous ICML reviewers for their constructive comments that helped improve the quality of this paper.

# Impact Statement

This paper presents work whose goal is to advance the field of machine learning. In particular, it develops a method for representing irregular multivariate time series so that existing time series backbones can better handle irregular observations. This may support applications in domains such as healthcare, climate science, and human activity analysis. While there are many potential societal consequences of this work, we do not identify any specific consequence that must be highlighted here.

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

# A    Dataset Details

|  | Human Activity | USHCN | PhysioNet | MIMIC-III |
|---|---|---|---|---|
| # Samples | 5400 | 26736 | 12000 | 23457 |
| # Variables | 12 | 5 | 36 | 96 |
| # Avg. Length | 120 | 163 | 74 | 46 |
| Missing Ratio | 75% | 77.9% | 88.4% | 96.7% |

*(a)* Forecasting

|  | P19 | P12 | PAM |
|---|---|---|---|
| # Samples | 38803 | 11988 | 5333 |
| # Variables | 34 | 36 | 17 |
| # Classes | 2 | 2 | 8 |
| Missing Ratio | 94.9% | 88.4% | 60.0% |

*(b)* Classification

*Table A.1.* Summary of irregular multivariate time series datasets for forecasting and classification tasks.

We evaluate our method on four forecasting benchmarks and three classification benchmarks. We follow the preprocessing pipeline of (Luo et al., 2025) for the forecasting datasets and the P12 and P19 classification datasets. For PAM, which is not covered in (Luo et al., 2025), we follow the preprocessing protocol of (Zhang et al., 2022).

## A.1    Forecasting

**Human Activity (Kaluža et al., 2010).**    The Human Activity dataset comprises 12 irregularly sampled three-dimensional positional variables recorded by wearable sensors attached to the ankles, belts, and chests of five individuals. The continuous sequences are divided into 5,400 irregular multivariate time series (IMTS) instances, each lasting 4,000 milliseconds. In each instance, the first 1,000, 2,000, or 3,000 milliseconds are observed to forecast the remaining 3,000, 2,000, or 1,000 milliseconds, respectively.

**USHCN (Menne et al., 2015).**    USHCN provides long-term climate records from weather stations across the United States, including five meteorological variables. After standard preprocessing, we extract data from 1,114 stations during 1996–2000, yielding 26,736 IMTS samples. For each sample, climate measurements from the preceding 24 months are used to predict conditions over the next 1, 6, or 12 months.

**PhysioNet (Silva et al., 2012).**    PhysioNet is a clinical benchmark dataset consisting of 12,000 IMTS instances from ICU patients. Each instance contains 36 physiological signals irregularly measured within the first 48 hours following admission. We construct three forecasting scenarios by using the initial 12, 24, or 36 hours as historical observations and predicting the remaining time span.

**MIMIC-III (Johnson et al., 2016).**    MIMIC-III is a large-scale critical care database including IMTS data from 23,457 patients with 96 clinical variables recorded during the first 48 hours of ICU stay. The same horizon configurations as those used for PhysioNet are applied.

## A.2    Classification

**P19 (Reyna et al., 2020).**    PhysioNet Sepsis Early Prediction Challenge 2019 Dataset contains 38,803 patient records with 34 temporal variables and static features such as age, gender, ICU type, and ICU stay information. Each sample has a binary label indicating whether sepsis will occur within the next 6 hours.

**P12 (Goldberger et al., 2000).**   PhysioNet Challenge 2012 Dataset includes 11,988 patient records after excluding 12 invalid samples identified by (Horn et al., 2020). Each sample consists of 36 physiological variables collected during the first 48 ICU hours and 9 static features including age and gender. Labels indicate whether ICU stay exceeds 3 days.

**PAM (Reiss & Stricker, 2012).**   Physical Activity Monitoring Dataset is an 8-class activity classification dataset derived from PAMAP2 after removing one subject and infrequent activities. It contains 5,333 samples with 600 observations each. To simulate irregular time series, 60% of observations are randomly removed.

# B   Baseline Details

## B.1   Methods for MTS

We evaluate QuITE across a diverse set of MTS backbones, including Transformer-, CNN-, MLP-, and Mamba-based architectures. Each baseline strictly adheres to the original hyperparameter configurations reported in their respective studies. For QuITE-integrated models, we employ a hidden dimension of 64 to ensure that performance gains are attributable to the method itself rather than increased model capacity. Since these backbones were primarily designed for forecasting, we adapt their original configurations for the classification task. The only modification is the hidden dimension, which is standardized to 64 to maintain consistency with the QuITE configuration.

**PatchTST**    (Nie et al., 2022) PatchTST is a Transformer-based model that segments time series into subseries-level patches and uses them as input tokens with a channel-independent design. In the standalone setting, we use 3 layers, 4 attention heads, and a hidden dimension of 256. When integrated with QuITE, the hidden dimension is reduced to 64 to align with our unified configuration.

**iTransformer**    (Liu et al., 2023) iTransformer rethinks the Transformer architecture by applying attention and feed-forward networks along inverted dimensions to better capture multivariate dependencies. In the standalone setting, we use 3 layers, 4 attention heads, and a hidden dimension of 512. When integrated with QuITE, the hidden dimension is reduced to 64.

**TimeXer**    (Wang et al., 2024) TimeXer jointly models endogenous and exogenous information through patch-wise self-attention and variate-wise cross-attention. In the standalone setting, we use 3 layers, 4 attention heads, and a hidden dimension of 256. When integrated with QuITE, the hidden dimension is reduced to 64.

**PatchMixer**    (Gong et al., 2023) PatchMixer is a CNN-based architecture that employs permutation-variant convolutions to preserve temporal ordering across patches. We use a single-layer architecture with a hidden dimension of 256 for the standalone baseline, and reduce the hidden dimension to 64 when QuITE is applied.

**TMix**    (Chen et al., 2023a) TMix, also known as TSMixer, stacks temporal linear layers with nonlinear activations to efficiently capture temporal dependencies. The standalone setting uses 2 layers with a hidden dimension of 128, while the QuITE-based setting uses a hidden dimension of 64. Since TMix is not originally designed as a patch-based model, we replace its input embedding with the patch embedding used in PatchTST to ensure compatibility with patch-based forecasting settings.

The above five models are implemented based on the official Time-Series-Library repository.[1]

**S-Mamba**    (Wang et al., 2025) S-Mamba is a state-space-based model built upon the Mamba architecture. It tokenizes each variate independently via linear projection, followed by bidirectional Mamba layers to capture inter-variable correlations and feed-forward networks for temporal dynamics. We use 2 layers with a hidden dimension of 256 in the standalone setting, and reduce the hidden dimension to 64 when QuITE is applied. We use the official implementation.[2]

## B.2   Methods for IMTS

For HyperIMTS (Li et al., 2025) and Hi-Patch (Luo et al., 2025), we reproduced the experiments using the official implementations and hyperparameter settings released by the authors. For all other baseline methods, the reported results were directly adopted from (Luo et al., 2025).

**Warpformer**    (Zhang et al., 2023) employs a tailored input encoding scheme that captures intra-series temporal irregularity as well as inter-series discrepancies. In addition, a warping module is introduced to adaptively align time series across different temporal scales, followed by a customized attention mechanism for effective representation learning.

**Raindrop**    (Zhang et al., 2022) formulates each multivariate time series sample as an individual sensor graph. By leveraging a novel message-passing operator, it dynamically models time-varying dependency structures among sensors.

---

[1] https://github.com/thuml/Time-Series-Library/tree/main/models
[2] https://github.com/wzhwzhwzh0921/S-D-Mamba

**GRU-D** (Che et al., 2018) incorporates missingness information into recurrent modeling. It utilizes both masking vectors and time interval information to represent missing patterns, enabling the model to jointly learn long-range temporal dependencies and informative missing-value dynamics.

**tPatchGNN** (Zhang et al., 2024) converts univariate irregular time series into sequences of flexible patches, where each patch contains a variable number of observations but maintains a uniform temporal resolution. This patch-based formulation allows the model to capture local temporal semantics and cross-series correlations.

**GraFITi** (Yalavarthi et al., 2024) reformulates time series forecasting as a graph learning problem. It first constructs a Sparsity Structure Graph, represented as a sparse bipartite graph, and then performs forecasting by predicting edge weights within this structure.

**CRU** (Schirmer et al., 2022) assumes an underlying latent state governed by a linear stochastic differential equation. Embedded in an encoder–decoder framework, the state evolution and update rules are analytically derived using continuous–discrete Kalman filtering.

**mTAND** (Shukla & Marlin, 2021) learns continuous-time embeddings of observed values and applies an attention-based aggregation mechanism to obtain fixed-dimensional representations from irregularly sampled time series.

**NeuralFlow** (Biloš et al., 2021) provides an alternative approach that represents ODE dynamics by learning the solution trajectories directly with a neural network.

**Latent-ODE** (Rubanova et al., 2019) extends conventional recurrent models by defining hidden-state dynamics in continuous time using ordinary differential equations.

**HyperIMTS** (Li et al., 2025) models IMTS using a hypergraph neural network. Each observation is represented as a node, and temporal as well as variable-level hyperedges are constructed to facilitate global message passing among all observed values. We use the official implementation.[3]

**Hi-Patch** (Luo et al., 2025) introduces a hierarchical patch-based graph modeling approach. Individual observations are first treated as nodes and encoded through an intra-patch graph layer to capture local temporal and inter-variable relationships. We use the official implementation.[4]

---

[3] https://github.com/Ladbaby/PyOmniTS
[4] https://github.com/qianlima-lab/Hi-Patch

# C   More Implementation Details

For both forecasting and classification tasks, we largely follow the experimental protocol of (Luo et al., 2025), including optimizer configurations, learning rates, batch sizes, and dataset-specific settings such as patch size and stride. Unless otherwise specified, we adopt the recommended settings from the original paper. Since PAM is not covered in (Luo et al., 2025), we follow the existing classification settings for the learning rate and batch size, and empirically select the patch size and stride. In addition to the protocol above, we use dataset-dependent settings for batch size, learning rate, and patching parameters, including patch size and stride, as summarized below.

| Dataset | Learning rate | Batch size | Patch / Stride | History length |
|---|---|---|---|---|
| Human Activity | $10^{-3}$ | 32 | 750 / 750
500 / 500
250 / 250 | 3000
2000
1000 |
| USHCN | $10^{-3}$ | 128 | 1.5 / 1.5 | 24 (default) |
| PhysioNet | $10^{-3}$ | 64 | 6 / 6
9 / 9
6 / 6 | 24
36
12 |
| MIMIC-III | $10^{-3}$ | 8
8
16 | 12 / 12
4.5 / 4.5
6 / 6 | 24
36
12 |

*(a)* Forecasting

| Dataset | Learning rate | Batch size | Patch / Stride |
|---|---|---|---|
| P19 | $10^{-3}$ | 64 | 3.75 / 3.75 |
| P12 | $10^{-3}$ | 64 | 6 / 6 |
| PAM | $10^{-3}$ | 64 | 10 / 10 |

*(b)* Classification

*Table C.1.* Training configurations for irregular multivariate time series datasets.

# D  More Results of Experiments

## D.1  Forecasting

| Human Activity | | | | | | |
|---|---|---|---|---|---|---|
| Horizon | 1000ms → 3000ms | | 2000ms → 2000ms | | 3000ms → 1000ms | |
| Metric | MSE $\times10^{-3}$ | MAE $\times10^{-2}$ | MSE $\times10^{-3}$ | MAE $\times10^{-2}$ | MSE $\times10^{-3}$ | MAE $\times10^{-2}$ |
| PatchTST | $5.06 \pm 0.04$ | $4.52 \pm 0.08$ | $4.02 \pm 0.04$ | $3.99 \pm 0.04$ | $3.10 \pm 0.12$ | $3.44 \pm 0.11$ |
| PatchMixer | $4.80 \pm 0.04$ | $4.40 \pm 0.07$ | $3.76 \pm 0.03$ | $3.84 \pm 0.02$ | $2.84 \pm 0.02$ | $3.21 \pm 0.03$ |
| TMix | $4.97 \pm 0.06$ | $4.48 \pm 0.06$ | $3.87 \pm 0.04$ | $3.92 \pm 0.04$ | $2.93 \pm 0.03$ | $3.28 \pm 0.02$ |
| iTransformer | $5.71 \pm 0.04$ | $5.20 \pm 0.05$ | $4.88 \pm 0.08$ | $4.80 \pm 0.03$ | $3.77 \pm 0.02$ | $4.13 \pm 0.02$ |
| S-Mamba | $5.37 \pm 0.05$ | $5.10 \pm 0.03$ | $4.68 \pm 0.04$ | $4.77 \pm 0.06$ | $3.56 \pm 0.04$ | $4.05 \pm 0.03$ |
| TimeXer | $4.77 \pm 0.07$ | $4.50 \pm 0.06$ | $3.87 \pm 0.07$ | $3.96 \pm 0.05$ | $2.99 \pm 0.03$ | $3.41 \pm 0.05$ |
| Warpformer | $4.26 \pm 0.11$ | $4.26 \pm 0.04$ | $3.60 \pm 0.08$ | $3.81 \pm 0.03$ | $2.61 \pm 0.02$ | $3.12 \pm 0.01$ |
| Raindrop | $5.75 \pm 0.33$ | $5.37 \pm 0.22$ | $5.57 \pm 0.34$ | $5.15 \pm 0.11$ | $4.42 \pm 0.25$ | $4.65 \pm 0.14$ |
| GRU-D | $6.14 \pm 0.76$ | $5.75 \pm 0.49$ | $5.93 \pm 0.10$ | $5.66 \pm 0.66$ | $3.94 \pm 0.29$ | $4.37 \pm 0.21$ |
| tPatchGNN | $4.56 \pm 0.08$ | $4.32 \pm 0.06$ | $3.71 \pm 0.20$ | $3.89 \pm 0.16$ | $2.79 \pm 0.09$ | $3.24 \pm 0.06$ |
| GraFITi | $4.91 \pm 0.07$ | $4.62 \pm 0.03$ | $4.59 \pm 0.04$ | $4.45 \pm 0.04$ | $3.03 \pm 0.14$ | $3.45 \pm 0.10$ |
| CRU | $4.85 \pm 0.09$ | $4.86 \pm 0.07$ | $4.12 \pm 0.08$ | $4.43 \pm 0.06$ | $3.03 \pm 0.04$ | $3.60 \pm 0.04$ |
| mTAND | $5.29 \pm 0.32$ | $5.12 \pm 0.23$ | $4.38 \pm 0.37$ | $4.59 \pm 0.29$ | $3.14 \pm 0.09$ | $3.71 \pm 0.06$ |
| NeuralFlow | $6.01 \pm 0.91$ | $5.66 \pm 0.60$ | $5.47 \pm 0.49$ | $5.35 \pm 0.28$ | $4.29 \pm 0.63$ | $4.61 \pm 0.43$ |
| Latent-ODE | $5.48 \pm 0.21$ | $5.33 \pm 0.14$ | $5.04 \pm 0.46$ | $5.11 \pm 0.29$ | $3.32 \pm 0.10$ | $3.91 \pm 0.08$ |
| HyperIMTS | $4.00 \pm 0.13$ | $4.13 \pm 0.10$ | $3.15 \pm 0.03$ | $3.58 \pm 0.05$ | $2.49 \pm 0.01$ | $3.02 \pm 0.01$ |
| Hi-Patch | $4.20 \pm 0.09$ | $4.22 \pm 0.05$ | $3.26 \pm 0.02$ | $3.67 \pm 0.03$ | $2.56 \pm 0.02$ | $3.12 \pm 0.02$ |
| PatchTST + QuITE | $4.69 \pm 0.01$ | $4.29 \pm 0.03$ | $3.62 \pm 0.02$ | $3.74 \pm 0.02$ | $2.76 \pm 0.01$ | $3.14 \pm 0.02$ |
| PatchMixer + QuITE | $4.71 \pm 0.02$ | $4.31 \pm 0.04$ | $3.67 \pm 0.02$ | $3.75 \pm 0.03$ | $2.78 \pm 0.02$ | $3.13 \pm 0.01$ |
| TMix + QuITE | $4.75 \pm 0.03$ | $4.38 \pm 0.05$ | $3.66 \pm 0.02$ | $3.74 \pm 0.02$ | $2.77 \pm 0.01$ | $3.15 \pm 0.02$ |
| iTransformer + QuITE | $4.10 \pm 0.08$ | $4.18 \pm 0.05$ | $3.25 \pm 0.06$ | $3.65 \pm 0.04$ | $2.58 \pm 0.02$ | $3.12 \pm 0.01$ |
| S-Mamba + QuITE | $4.20 \pm 0.09$ | $4.22 \pm 0.04$ | $3.37 \pm 0.02$ | $3.72 \pm 0.02$ | $2.72 \pm 0.03$ | $3.24 \pm 0.04$ |
| TimeXer + QuITE | $4.04 \pm 0.05$ | $4.03 \pm 0.01$ | $3.19 \pm 0.04$ | $3.57 \pm 0.03$ | $2.53 \pm 0.02$ | $3.04 \pm 0.02$ |
| QuITE$^{++}$ | $3.96 \pm 0.04$ | $4.04 \pm 0.03$ | $3.11 \pm 0.03$ | $3.49 \pm 0.04$ | $2.46 \pm 0.02$ | $2.92 \pm 0.06$ |

*Table D.1.1.* Results on the Human Activity dataset. The results are reported as (Mean $\pm$ Std).

| USHCN | | | | | | |
|---|---|---|---|---|---|---|
| Horizon | 24months → 1months | | 24months → 6months | | 24months → 12months | |
| Metric | MSE $\times 10^{-1}$ | MAE $\times 10^{-1}$ | MSE $\times 10^{-1}$ | MAE $\times 10^{-1}$ | MSE $\times 10^{-1}$ | MAE $\times 10^{-1}$ |
| PatchTST | $5.35 \pm 0.07$ | $3.31 \pm 0.26$ | $5.30 \pm 0.08$ | $3.32 \pm 0.27$ | $5.11 \pm 0.06$ | $3.09 \pm 0.07$ |
| PatchMixer | $5.22 \pm 0.05$ | $3.14 \pm 0.10$ | $5.31 \pm 0.12$ | $3.13 \pm 0.12$ | $5.22 \pm 0.05$ | $3.12 \pm 0.06$ |
| TMix | $5.41 \pm 0.12$ | $3.27 \pm 0.07$ | $7.31 \pm 0.03$ | $4.49 \pm 0.14$ | $7.53 \pm 0.01$ | $4.56 \pm 0.12$ |
| iTransformer | $5.91 \pm 0.10$ | $3.63 \pm 0.09$ | $5.51 \pm 0.11$ | $3.22 \pm 0.20$ | $5.46 \pm 0.12$ | $3.21 \pm 0.10$ |
| S-Mamba | $5.73 \pm 0.12$ | $3.57 \pm 0.20$ | $5.50 \pm 0.11$ | $3.25 \pm 0.10$ | $5.40 \pm 0.06$ | $3.20 \pm 0.02$ |
| TimeXer | $5.36 \pm 0.20$ | $3.24 \pm 0.16$ | $5.21 \pm 0.10$ | $3.17 \pm 0.13$ | $5.23 \pm 0.12$ | $3.07 \pm 0.09$ |
| Warpformer | $5.09 \pm 0.03$ | $3.10 \pm 0.04$ | $5.12 \pm 0.03$ | $3.13 \pm 0.08$ | $5.10 \pm 0.07$ | $3.13 \pm 0.12$ |
| Raindrop | $5.64 \pm 0.10$ | $3.29 \pm 0.03$ | $7.01 \pm 0.49$ | $4.24 \pm 0.33$ | $7.61 \pm 0.02$ | $4.61 \pm 0.05$ |
| GRU-D | $5.17 \pm 0.06$ | $3.21 \pm 0.05$ | $5.29 \pm 0.09$ | $3.34 \pm 0.09$ | $5.36 \pm 0.12$ | $3.25 \pm 0.07$ |
| tPatchGNN | $5.00 \pm 0.03$ | $3.07 \pm 0.05$ | $5.23 \pm 0.02$ | $3.24 \pm 0.19$ | $6.23 \pm 0.10$ | $3.83 \pm 0.60$ |
| GraFITi | $5.07 \pm 0.03$ | $2.97 \pm 0.04$ | $5.12 \pm 0.14$ | $3.09 \pm 0.10$ | $5.01 \pm 0.03$ | $3.14 \pm 0.06$ |
| CRU | $5.15 \pm 0.50$ | $3.18 \pm 0.03$ | $6.77 \pm 1.04$ | $4.11 \pm 0.61$ | $6.64 \pm 0.95$ | $4.08 \pm 0.51$ |
| mTAND | $5.03 \pm 0.05$ | $3.00 \pm 0.06$ | $5.16 \pm 0.10$ | $3.10 \pm 0.07$ | $5.07 \pm 0.03$ | $3.09 \pm 0.03$ |
| NeuralFlow | $5.41 \pm 0.05$ | $3.35 \pm 0.06$ | $5.52 \pm 0.05$ | $3.46 \pm 0.05$ | $5.48 \pm 0.37$ | $3.56 \pm 0.37$ |
| Latent-ODE | $5.16 \pm 0.04$ | $3.21 \pm 0.07$ | $5.18 \pm 0.04$ | $3.36 \pm 0.04$ | $5.23 \pm 0.04$ | $3.35 \pm 0.02$ |
| HyperIMTS | $4.96 \pm 0.06$ | $3.00 \pm 0.10$ | $4.99 \pm 0.19$ | $3.10 \pm 0.18$ | $4.97 \pm 0.16$ | $3.08 \pm 0.08$ |
| Hi-Patch | $5.00 \pm 0.09$ | $3.03 \pm 0.05$ | $5.13 \pm 0.11$ | $3.05 \pm 0.11$ | $5.04 \pm 0.04$ | $3.03 \pm 0.04$ |
| PatchTST + QuITE | $5.07 \pm 0.04$ | $3.01 \pm 0.07$ | $5.06 \pm 0.06$ | $3.17 \pm 0.07$ | $5.04 \pm 0.06$ | $3.00 \pm 0.04$ |
| PatchMixer + QuITE | $5.11 \pm 0.05$ | $3.06 \pm 0.11$ | $5.02 \pm 0.03$ | $3.04 \pm 0.10$ | $5.00 \pm 0.02$ | $3.07 \pm 0.10$ |
| TMix + QuITE | $5.18 \pm 0.13$ | $3.10 \pm 0.17$ | $5.52 \pm 0.98$ | $3.39 \pm 0.77$ | $5.03 \pm 0.04$ | $3.05 \pm 0.08$ |
| iTransformer + QuITE | $5.06 \pm 0.04$ | $3.06 \pm 0.06$ | $4.86 \pm 0.09$ | $2.96 \pm 0.10$ | $4.94 \pm 0.06$ | $3.01 \pm 0.06$ |
| S-Mamba + QuITE | $5.04 \pm 0.05$ | $3.06 \pm 0.08$ | $4.93 \pm 0.05$ | $3.04 \pm 0.05$ | $4.93 \pm 0.09$ | $3.02 \pm 0.07$ |
| TimeXer + QuITE | $4.97 \pm 0.04$ | $2.97 \pm 0.06$ | $4.98 \pm 0.06$ | $3.05 \pm 0.07$ | $4.93 \pm 0.06$ | $2.97 \pm 0.07$ |
| QuITE$^{++}$ | $4.84 \pm 0.02$ | $2.92 \pm 0.06$ | $4.81 \pm 0.04$ | $2.94 \pm 0.04$ | $4.81 \pm 0.05$ | $2.93 \pm 0.03$ |

*Table D.1.2.* Results on the USHCN dataset. The results are reported as (Mean $\pm$ Std).

| PhysioNet | | | | | | |
|---|---|---|---|---|---|---|
| Horizon | 12h → 36h | | 24h → 24h | | 36h → 12h | |
| Metric | MSE $\times10^{-2}$ | MAE $\times10^{-2}$ | MSE $\times10^{-2}$ | MAE $\times10^{-2}$ | MSE $\times10^{-2}$ | MAE $\times10^{-2}$ |
| PatchTST | $20.03 \pm 0.30$ | $8.16 \pm 0.09$ | $15.28 \pm 0.89$ | $7.02 \pm 0.45$ | $12.97 \pm 0.33$ | $6.07 \pm 0.09$ |
| PatchMixer | $18.62 \pm 0.10$ | $7.64 \pm 0.06$ | $12.28 \pm 0.32$ | $5.88 \pm 0.15$ | $10.33 \pm 0.20$ | $5.03 \pm 0.06$ |
| TMix | $18.57 \pm 0.05$ | $7.65 \pm 0.04$ | $12.38 \pm 0.07$ | $5.81 \pm 0.07$ | $10.50 \pm 0.08$ | $5.13 \pm 0.03$ |
| iTransformer | $21.14 \pm 0.51$ | $8.74 \pm 0.35$ | $16.48 \pm 0.12$ | $7.94 \pm 0.19$ | $14.09 \pm 0.28$ | $6.68 \pm 0.23$ |
| S-Mamba | $8.23 \pm 0.22$ | $4.71 \pm 0.06$ | $6.93 \pm 0.24$ | $4.40 \pm 0.19$ | $6.26 \pm 0.48$ | $4.25 \pm 0.14$ |
| TimeXer | $6.91 \pm 0.08$ | $4.43 \pm 0.02$ | $5.79 \pm 0.09$ | $4.14 \pm 0.07$ | $4.94 \pm 0.22$ | $3.72 \pm 0.11$ |
| Warpformer | $6.51 \pm 0.12$ | $4.24 \pm 0.04$ | $5.04 \pm 0.14$ | $3.72 \pm 0.06$ | $4.17 \pm 0.13$ | $3.38 \pm 0.08$ |
| Raindrop | $10.24 \pm 0.18$ | $5.83 \pm 0.10$ | $10.63 \pm 0.29$ | $6.02 \pm 0.19$ | $10.67 \pm 0.33$ | $5.87 \pm 0.20$ |
| GRU-D | $7.80 \pm 0.22$ | $5.13 \pm 0.13$ | $5.76 \pm 0.34$ | $4.53 \pm 0.15$ | $6.85 \pm 0.37$ | $4.88 \pm 0.18$ |
| tPatchGNN | $6.45 \pm 0.11$ | $4.24 \pm 0.09$ | $5.06 \pm 0.10$ | $3.75 \pm 0.07$ | $4.22 \pm 0.09$ | $3.38 \pm 0.04$ |
| GraFITi | $6.30 \pm 0.14$ | $4.38 \pm 0.12$ | $5.11 \pm 0.19$ | $3.96 \pm 0.09$ | $4.58 \pm 0.11$ | $3.65 \pm 0.05$ |
| CRU | $7.66 \pm 0.14$ | $4.97 \pm 0.05$ | $6.43 \pm 0.62$ | $4.51 \pm 0.16$ | $6.74 \pm 0.21$ | $4.82 \pm 0.11$ |
| mTAND | $7.46 \pm 0.19$ | $4.85 \pm 0.05$ | $6.18 \pm 0.31$ | $4.44 \pm 0.19$ | $5.61 \pm 0.31$ | $4.15 \pm 0.09$ |
| NeuralFlow | $7.98 \pm 0.57$ | $5.08 \pm 0.24$ | $7.68 \pm 0.37$ | $4.84 \pm 0.19$ | $8.87 \pm 1.00$ | $5.43 \pm 0.18$ |
| Latent-ODE | $7.28 \pm 0.13$ | $4.83 \pm 0.07$ | $6.85 \pm 0.28$ | $4.77 \pm 0.17$ | $6.99 \pm 0.24$ | $4.74 \pm 0.11$ |
| HyperIMTS | $6.11 \pm 0.16$ | $4.23 \pm 0.16$ | $4.65 \pm 0.14$ | $3.56 \pm 0.09$ | $3.99 \pm 0.04$ | $3.21 \pm 0.04$ |
| Hi-Patch | $6.39 \pm 0.12$ | $4.10 \pm 0.05$ | $5.07 \pm 0.22$ | $3.63 \pm 0.03$ | $4.27 \pm 0.15$ | $3.30 \pm 0.06$ |
| PatchTST + QuITE | $17.47 \pm 0.04$ | $7.15 \pm 0.04$ | $10.62 \pm 0.05$ | $5.17 \pm 0.06$ | $8.87 \pm 0.07$ | $4.43 \pm 0.05$ |
| PatchMixer + QuITE | $17.52 \pm 0.04$ | $7.22 \pm 0.05$ | $11.88 \pm 1.31$ | $5.62 \pm 0.45$ | $9.06 \pm 0.25$ | $4.55 \pm 0.16$ |
| TMix + QuITE | $17.48 \pm 0.02$ | $7.21 \pm 0.02$ | $10.72 \pm 0.09$ | $5.22 \pm 0.07$ | $8.96 \pm 0.18$ | $4.50 \pm 0.10$ |
| iTransformer + QuITE | $6.32 \pm 0.15$ | $4.15 \pm 0.07$ | $4.99 \pm 0.23$ | $3.65 \pm 0.08$ | $4.33 \pm 0.20$ | $3.34 \pm 0.10$ |
| S-Mamba + QuITE | $6.26 \pm 0.09$ | $4.11 \pm 0.09$ | $5.11 \pm 0.19$ | $3.67 \pm 0.06$ | $4.11 \pm 0.19$ | $3.27 \pm 0.12$ |
| TimeXer + QuITE | $6.18 \pm 0.10$ | $4.08 \pm 0.12$ | $4.91 \pm 0.30$ | $3.64 \pm 0.11$ | $4.06 \pm 0.06$ | $3.27 \pm 0.06$ |
| QuITE$^{++}$ | $6.08 \pm 0.14$ | $3.99 \pm 0.08$ | $4.99 \pm 0.18$ | $3.62 \pm 0.07$ | $3.81 \pm 0.12$ | $3.18 \pm 0.07$ |

*Table D.1.3.* Results on the PhysioNet dataset. The results are reported as (Mean $\pm$ Std).

| MIMIC-III | | | | | | |
|---|---|---|---|---|---|---|
| Horizon | 12h → 36h | | 24h → 24h | | 36h → 12h | |
| Metric | MSE $\times 10^{-3}$ | MAE $\times 10^{-2}$ | MSE $\times 10^{-3}$ | MAE $\times 10^{-2}$ | MSE $\times 10^{-3}$ | MAE $\times 10^{-2}$ |
| PatchTST | $5.67 \pm 0.12$ | $17.62 \pm 0.28$ | $4.33 \pm 0.03$ | $13.28 \pm 0.07$ | $4.87 \pm 0.22$ | $14.12 \pm 0.85$ |
| PatchMixer | $5.53 \pm 0.04$ | $17.23 \pm 0.13$ | $4.10 \pm 0.01$ | $12.62 \pm 0.14$ | $4.11 \pm 0.04$ | $12.10 \pm 0.05$ |
| TMix | $5.57 \pm 0.01$ | $17.30 \pm 0.18$ | $4.14 \pm 0.02$ | $12.77 \pm 0.20$ | $4.04 \pm 0.09$ | $12.05 \pm 0.41$ |
| iTransformer | $6.00 \pm 0.07$ | $19.06 \pm 0.44$ | $5.50 \pm 0.69$ | $17.20 \pm 1.98$ | $6.05 \pm 0.76$ | $17.84 \pm 2.71$ |
| S-Mamba | $2.83 \pm 0.43$ | $10.89 \pm 1.05$ | $2.38 \pm 0.05$ | $9.38 \pm 0.23$ | $3.92 \pm 0.15$ | $11.74 \pm 0.64$ |
| TimeXer | $2.01 \pm 0.03$ | $8.44 \pm 0.10$ | $1.92 \pm 0.05$ | $7.98 \pm 0.23$ | $1.98 \pm 0.12$ | $8.64 \pm 0.69$ |
| Warpformer | $2.32 \pm 0.04$ | $8.14 \pm 0.07$ | $1.76 \pm 0.30$ | $7.27 \pm 0.15$ | $1.45 \pm 0.10$ | $6.74 \pm 0.08$ |
| Raindrop | $2.36 \pm 0.03$ | $8.63 \pm 0.11$ | $2.31 \pm 0.07$ | $8.61 \pm 0.12$ | $2.21 \pm 0.37$ | $9.17 \pm 0.49$ |
| GRU-D | $2.39 \pm 0.02$ | $8.43 \pm 0.13$ | $2.35 \pm 0.06$ | $8.34 \pm 0.22$ | $2.03 \pm 0.13$ | $8.14 \pm 0.26$ |
| tPatchGNN | $2.35 \pm 0.03$ | $8.23 \pm 0.08$ | $1.97 \pm 0.05$ | $7.76 \pm 0.22$ | $1.44 \pm 0.08$ | $6.78 \pm 0.14$ |
| GraFITi | $2.22 \pm 0.05$ | $8.13 \pm 0.13$ | $1.76 \pm 0.04$ | $7.28 \pm 0.13$ | $1.61 \pm 0.27$ | $7.16 \pm 0.36$ |
| CRU | $2.34 \pm 0.05$ | $8.32 \pm 0.13$ | $2.23 \pm 0.03$ | $7.99 \pm 0.22$ | $2.00 \pm 0.13$ | $8.16 \pm 0.26$ |
| mTAND | $2.29 \pm 0.03$ | $8.38 \pm 0.08$ | $2.15 \pm 0.05$ | $8.00 \pm 0.06$ | $2.01 \pm 0.09$ | $8.13 \pm 0.23$ |
| NeuralFlow | $2.26 \pm 0.08$ | $8.29 \pm 0.10$ | $2.34 \pm 0.05$ | $8.09 \pm 0.09$ | $1.97 \pm 0.12$ | $8.39 \pm 0.25$ |
| Latent-ODE | $2.38 \pm 0.05$ | $8.35 \pm 0.13$ | $2.11 \pm 0.15$ | $7.76 \pm 0.08$ | $1.90 \pm 0.03$ | $7.92 \pm 0.17$ |
| HyperIMTS | $1.85 \pm 0.02$ | $7.71 \pm 0.07$ | $1.68 \pm 0.01$ | $6.92 \pm 0.03$ | $1.52 \pm 0.08$ | $6.68 \pm 0.02$ |
| Hi-Patch | $1.88 \pm 0.01$ | $7.95 \pm 0.10$ | $1.70 \pm 0.01$ | $7.18 \pm 0.03$ | $1.56 \pm 0.02$ | $6.74 \pm 0.06$ |
| PatchTST + QuITE | $5.37 \pm 0.02$ | $17.01 \pm 0.09$ | $3.90 \pm 0.03$ | $12.35 \pm 0.12$ | $3.82 \pm 0.02$ | $11.84 \pm 0.27$ |
| PatchMixer + QuITE | $5.37 \pm 0.01$ | $16.96 \pm 0.11$ | $3.94 \pm 0.02$ | $12.34 \pm 0.20$ | $3.87 \pm 0.02$ | $11.51 \pm 0.28$ |
| TMix + QuITE | $5.41 \pm 0.05$ | $16.94 \pm 0.10$ | $3.98 \pm 0.02$ | $12.32 \pm 0.09$ | $3.87 \pm 0.03$ | $11.63 \pm 0.23$ |
| iTransformer + QuITE | $1.83 \pm 0.04$ | $7.64 \pm 0.05$ | $1.67 \pm 0.07$ | $6.93 \pm 0.14$ | $1.56 \pm 0.03$ | $6.78 \pm 0.09$ |
| S-Mamba + QuITE | $1.82 \pm 0.04$ | $7.56 \pm 0.12$ | $1.64 \pm 0.02$ | $6.90 \pm 0.07$ | $1.52 \pm 0.02$ | $6.64 \pm 0.13$ |
| TimeXer + QuITE | $1.84 \pm 0.03$ | $7.67 \pm 0.11$ | $1.68 \pm 0.03$ | $7.12 \pm 0.14$ | $1.55 \pm 0.03$ | $6.73 \pm 0.15$ |
| QuITE$^{++}$ | $1.80 \pm 0.02$ | $7.54 \pm 0.06$ | $1.63 \pm 0.04$ | $6.83 \pm 0.08$ | $1.48 \pm 0.02$ | $6.56 \pm 0.06$ |

*Table D.1.4.* Results on the MIMIC-III dataset. The results are reported as (Mean $\pm$ Std).

## D.2 Classification

| Dataset | Metric | Patch | | | | | |
|---------|--------|-------|------|-----------|------|------|------|
| | | **PatchTST** | + QuITE | **PatchMixer** | + QuITE | **TMix** | + QuITE |
| P12 | AUROC | $83.6 \pm 0.7$ | $84.5 \pm 0.9$ | $78.2 \pm 3.2$ | $83.9 \pm 2.6$ | $82.8 \pm 1.7$ | $85.2 \pm 1.0$ |
| | AUPRC | $47.0 \pm 3.0$ | $54.5 \pm 2.6$ | $39.2 \pm 2.7$ | $45.2 \pm 2.8$ | $45.5 \pm 3.4$ | $51.9 \pm 4.3$ |
| P19 | AUROC | $82.4 \pm 2.6$ | $85.0 \pm 1.8$ | $75.0 \pm 3.0$ | $83.8 \pm 2.7$ | $63.0 \pm 10.2$ | $81.2 \pm 6.3$ |
| | AUPRC | $40.3 \pm 2.7$ | $54.5 \pm 2.1$ | $26.4 \pm 2.5$ | $55.8 \pm 2.0$ | $10.1 \pm 5.9$ | $38.0 \pm 4.3$ |
| PAM | Accuracy | $84.7 \pm 1.3$ | $88.1 \pm 1.4$ | $73.5 \pm 3.5$ | $82.2 \pm 5.4$ | $80.0 \pm 1.7$ | $84.3 \pm 1.3$ |
| | Precision | $86.4 \pm 1.6$ | $89.6 \pm 1.7$ | $77.6 \pm 2.4$ | $86.4 \pm 3.5$ | $82.0 \pm 1.7$ | $86.1 \pm 1.4$ |
| | Recall | $86.5 \pm 1.0$ | $89.3 \pm 1.2$ | $75.6 \pm 4.7$ | $82.7 \pm 6.6$ | $81.3 \pm 1.5$ | $86.1 \pm 1.3$ |
| | F1 score | $86.3 \pm 1.2$ | $89.3 \pm 1.4$ | $75.7 \pm 3.5$ | $83.7 \pm 5.7$ | $81.4 \pm 1.4$ | $85.9 \pm 1.3$ |

| Dataset | Metric | Variate | | | | Hybrid | |
|---------|--------|---------|------|----------|------|--------|------|
| | | **iTransformer** | + QuITE | **S-Mamba** | + QuITE | **TimeXer** | + QuITE |
| P12 | AUROC | $82.6 \pm 0.7$ | $85.3 \pm 1.0$ | $82.3 \pm 0.7$ | $84.1 \pm 1.6$ | $83.5 \pm 2.0$ | $86.1 \pm 0.5$ |
| | AUPRC | $45.3 \pm 3.1$ | $47.6 \pm 2.0$ | $43.6 \pm 1.8$ | $49.5 \pm 4.1$ | $48.0 \pm 3.4$ | $49.5 \pm 2.1$ |
| P19 | AUROC | $82.5 \pm 1.8$ | $86.5 \pm 3.0$ | $80.2 \pm 1.3$ | $85.4 \pm 3.3$ | $83.2 \pm 2.4$ | $85.3 \pm 1.4$ |
| | AUPRC | $39.2 \pm 4.5$ | $51.7 \pm 5.5$ | $37.0 \pm 6.7$ | $51.7 \pm 4.0$ | $41.9 \pm 4.3$ | $54.9 \pm 0.9$ |
| PAM | Accuracy | $87.6 \pm 1.4$ | $90.7 \pm 1.2$ | $85.5 \pm 1.7$ | $87.4 \pm 1.1$ | $85.6 \pm 1.5$ | $89.4 \pm 0.7$ |
| | Precision | $89.3 \pm 1.5$ | $91.8 \pm 1.3$ | $86.1 \pm 1.6$ | $89.0 \pm 1.1$ | $87.0 \pm 1.4$ | $90.2 \pm 1.1$ |
| | Recall | $89.3 \pm 1.2$ | $91.4 \pm 1.2$ | $85.7 \pm 1.2$ | $88.7 \pm 1.5$ | $87.6 \pm 1.7$ | $90.5 \pm 0.6$ |
| | F1 score | $89.2 \pm 1.2$ | $91.5 \pm 1.2$ | $85.9 \pm 1.5$ | $88.7 \pm 1.3$ | $87.0 \pm 1.4$ | $90.3 \pm 0.8$ |

*Table D.2.1.* Results on the classification dataset. The results are reported as (Mean ± Std).

## D.3 Forecasting Performance of Different Embedding Methods

| Dataset | Horizon | + Add MSE | + Add MAE | + Concat MSE | + Concat MAE | + mTAND MSE | + mTAND MAE | + Mean Pooling MSE | + Mean Pooling MAE | + QuITE MSE | + QuITE MAE |
|---|---|---|---|---|---|---|---|---|---|---|---|
| Human Activity (ms) | $3000 \to 1000$ | $3.10 \pm 0.06$ | $3.47 \pm 0.04$ | $2.90 \pm 0.03$ | $3.31 \pm 0.06$ | $2.76 \pm 0.01$ | $3.12 \pm 0.01$ | $2.77 \pm 0.01$ | $3.14 \pm 0.02$ | $2.76 \pm 0.01$ | $3.14 \pm 0.02$ |
| | $2000 \to 2000$ | $3.97 \pm 0.07$ | $4.03 \pm 0.06$ | $3.88 \pm 0.02$ | $3.91 \pm 0.03$ | $3.74 \pm 0.07$ | $3.77 \pm 0.05$ | $3.75 \pm 0.03$ | $3.80 \pm 0.04$ | $3.62 \pm 0.02$ | $3.74 \pm 0.02$ |
| | $1000 \to 3000$ | $4.92 \pm 0.11$ | $4.58 \pm 0.12$ | $4.92 \pm 0.07$ | $4.51 \pm 0.10$ | $4.73 \pm 0.08$ | $4.38 \pm 0.06$ | $4.72 \pm 0.05$ | $4.38 \pm 0.05$ | $4.69 \pm 0.01$ | $4.29 \pm 0.03$ |
| USHCN (m) | $24 \to 1$ | $5.33 \pm 0.09$ | $3.22 \pm 0.09$ | $5.32 \pm 0.05$ | $3.34 \pm 0.10$ | $5.17 \pm 0.02$ | $3.17 \pm 0.09$ | $5.16 \pm 0.05$ | $3.09 \pm 0.07$ | $5.07 \pm 0.04$ | $3.01 \pm 0.07$ |
| | $24 \to 6$ | $5.17 \pm 0.07$ | $3.11 \pm 0.16$ | $5.21 \pm 0.09$ | $3.15 \pm 0.07$ | $5.27 \pm 0.09$ | $3.24 \pm 0.13$ | $5.15 \pm 0.12$ | $3.25 \pm 0.23$ | $5.06 \pm 0.06$ | $3.17 \pm 0.07$ |
| | $24 \to 12$ | $5.19 \pm 0.13$ | $3.17 \pm 0.22$ | $5.12 \pm 0.06$ | $3.04 \pm 0.08$ | $5.19 \pm 0.07$ | $3.18 \pm 0.09$ | $5.10 \pm 0.03$ | $3.22 \pm 0.15$ | $5.04 \pm 0.06$ | $3.00 \pm 0.04$ |
| PhysioNet (h) | $12 \to 36$ | $18.44 \pm 0.15$ | $7.81 \pm 0.08$ | $18.13 \pm 0.19$ | $7.52 \pm 0.19$ | $18.53 \pm 0.15$ | $7.63 \pm 0.06$ | $18.08 \pm 0.43$ | $7.56 \pm 0.14$ | $17.47 \pm 0.04$ | $7.15 \pm 0.04$ |
| | $24 \to 24$ | $12.22 \pm 0.24$ | $6.08 \pm 0.10$ | $11.28 \pm 0.07$ | $5.49 \pm 0.08$ | $11.93 \pm 0.21$ | $5.64 \pm 0.11$ | $10.96 \pm 0.13$ | $5.38 \pm 0.16$ | $10.62 \pm 0.05$ | $5.17 \pm 0.06$ |
| | $36 \to 12$ | $10.70 \pm 0.22$ | $5.71 \pm 0.15$ | $9.50 \pm 0.05$ | $4.87 \pm 0.03$ | $9.68 \pm 0.04$ | $4.81 \pm 0.04$ | $9.47 \pm 0.13$ | $4.92 \pm 0.15$ | $8.87 \pm 0.07$ | $4.43 \pm 0.05$ |
| MIMIC-III (h) | $12 \to 36$ | $5.58 \pm 0.01$ | $17.44 \pm 0.15$ | $5.53 \pm 0.06$ | $17.37 \pm 0.38$ | $5.39 \pm 0.01$ | $17.11 \pm 0.06$ | $5.43 \pm 0.02$ | $17.17 \pm 0.10$ | $5.37 \pm 0.02$ | $17.01 \pm 0.09$ |
| | $24 \to 24$ | $4.40 \pm 0.02$ | $14.13 \pm 0.15$ | $4.06 \pm 0.02$ | $12.74 \pm 0.18$ | $3.93 \pm 0.01$ | $12.50 \pm 0.16$ | $4.02 \pm 0.03$ | $12.60 \pm 0.13$ | $3.90 \pm 0.03$ | $12.35 \pm 0.12$ |
| | $36 \to 12$ | $4.14 \pm 0.07$ | $12.66 \pm 0.28$ | $3.96 \pm 0.07$ | $12.28 \pm 0.69$ | $3.86 \pm 0.01$ | $11.80 \pm 0.35$ | $3.85 \pm 0.01$ | $11.74 \pm 0.52$ | $3.82 \pm 0.02$ | $11.84 \pm 0.27$ |

*Table D.3.1.* Forecasting performance of embedding variants with PatchTST.

| Dataset | Horizon | + Add MSE | + Add MAE | + Concat MSE | + Concat MAE | + mTAND MSE | + mTAND MAE | + Mean Pooling MSE | + Mean Pooling MAE | + QuITE MSE | + QuITE MAE |
|---|---|---|---|---|---|---|---|---|---|---|---|
| Human Activity (ms) | $3000 \to 1000$ | $3.98 \pm 0.09$ | $4.30 \pm 0.05$ | $6.49 \pm 5.66$ | $5.48 \pm 2.70$ | $2.67 \pm 0.03$ | $3.13 \pm 0.06$ | $2.70 \pm 0.01$ | $3.13 \pm 0.02$ | $2.58 \pm 0.02$ | $3.12 \pm 0.01$ |
| | $2000 \to 2000$ | $5.22 \pm 0.27$ | $4.99 \pm 0.15$ | $4.98 \pm 0.12$ | $4.87 \pm 0.09$ | $3.44 \pm 0.03$ | $3.65 \pm 0.03$ | $3.53 \pm 0.03$ | $3.73 \pm 0.03$ | $3.25 \pm 0.06$ | $3.65 \pm 0.04$ |
| | $1000 \to 3000$ | $5.74 \pm 0.05$ | $5.24 \pm 0.02$ | $5.83 \pm 0.18$ | $5.28 \pm 0.11$ | $4.38 \pm 0.06$ | $4.16 \pm 0.03$ | $4.55 \pm 0.03$ | $4.29 \pm 0.05$ | $4.10 \pm 0.08$ | $4.18 \pm 0.05$ |
| USHCN (m) | $24 \to 1$ | $5.85 \pm 0.18$ | $3.59 \pm 0.06$ | $5.81 \pm 0.06$ | $3.56 \pm 0.06$ | $5.31 \pm 0.04$ | $3.22 \pm 0.04$ | $5.14 \pm 0.04$ | $3.21 \pm 0.09$ | $5.06 \pm 0.04$ | $3.06 \pm 0.06$ |
| | $24 \to 6$ | $6.22 \pm 1.02$ | $3.69 \pm 0.74$ | $5.77 \pm 0.85$ | $3.47 \pm 0.52$ | $5.21 \pm 0.06$ | $3.10 \pm 0.06$ | $4.98 \pm 0.04$ | $3.02 \pm 0.06$ | $4.86 \pm 0.09$ | $2.96 \pm 0.10$ |
| | $24 \to 12$ | $6.72 \pm 1.18$ | $4.10 \pm 0.81$ | $6.70 \pm 1.19$ | $3.99 \pm 0.79$ | $5.17 \pm 0.06$ | $3.13 \pm 0.11$ | $5.12 \pm 0.06$ | $3.09 \pm 0.04$ | $4.94 \pm 0.06$ | $3.01 \pm 0.06$ |
| PhysioNet (h) | $12 \to 36$ | $15.66 \pm 0.18$ | $7.09 \pm 0.20$ | $15.81 \pm 0.25$ | $7.18 \pm 0.13$ | $18.13 \pm 0.19$ | $7.56 \pm 0.03$ | $16.76 \pm 0.06$ | $7.10 \pm 0.04$ | $6.32 \pm 0.15$ | $4.15 \pm 0.07$ |
| | $24 \to 24$ | $22.01 \pm 0.44$ | $9.03 \pm 0.16$ | $21.90 \pm 0.29$ | $8.88 \pm 0.26$ | $11.60 \pm 0.05$ | $5.52 \pm 0.05$ | $10.58 \pm 0.33$ | $5.35 \pm 0.19$ | $4.99 \pm 0.23$ | $3.65 \pm 0.08$ |
| | $36 \to 12$ | $17.10 \pm 0.14$ | $7.92 \pm 0.27$ | $17.10 \pm 0.25$ | $7.97 \pm 0.10$ | $9.60 \pm 0.08$ | $4.81 \pm 0.07$ | $9.11 \pm 0.11$ | $4.66 \pm 0.09$ | $4.33 \pm 0.20$ | $3.34 \pm 0.10$ |
| MIMIC-III (h) | $12 \to 36$ | $6.47 \pm 0.46$ | $19.99 \pm 1.60$ | $6.46 \pm 0.47$ | $20.07 \pm 1.28$ | $5.34 \pm 0.02$ | $16.94 \pm 0.08$ | $5.32 \pm 0.02$ | $17.01 \pm 0.04$ | $1.83 \pm 0.04$ | $7.64 \pm 0.05$ |
| | $24 \to 24$ | $6.47 \pm 0.26$ | $20.26 \pm 0.88$ | $6.60 \pm 0.00$ | $20.54 \pm 0.04$ | $3.83 \pm 0.04$ | $12.32 \pm 0.13$ | $3.86 \pm 0.02$ | $12.21 \pm 0.18$ | $1.67 \pm 0.07$ | $6.93 \pm 0.14$ |
| | $36 \to 12$ | $6.07 \pm 0.05$ | $18.93 \pm 0.40$ | $6.39 \pm 0.31$ | $20.24 \pm 1.18$ | $3.84 \pm 0.01$ | $11.89 \pm 0.27$ | $3.82 \pm 0.03$ | $11.62 \pm 0.16$ | $1.56 \pm 0.03$ | $6.78 \pm 0.09$ |

*Table D.3.2.* Forecasting performance of embedding variants with iTransformer.

| Dataset | Horizon | + Add MSE | + Add MAE | + Concat MSE | + Concat MAE | + mTAND MSE | + mTAND MAE | + Mean Pooling MSE | + Mean Pooling MAE | + QuITE MSE | + QuITE MAE |
|---|---|---|---|---|---|---|---|---|---|---|---|
| Human Activity (ms) | $3000 \to 1000$ | $2.73 \pm 0.02$ | $3.29 \pm 0.03$ | $2.61 \pm 0.01$ | $3.20 \pm 0.04$ | $2.67 \pm 0.02$ | $3.15 \pm 0.01$ | $2.56 \pm 0.03$ | $3.08 \pm 0.04$ | $2.46 \pm 0.02$ | $2.92 \pm 0.06$ |
| | $2000 \to 2000$ | $3.36 \pm 0.06$ | $3.73 \pm 0.04$ | $3.38 \pm 0.03$ | $3.74 \pm 0.04$ | $3.35 \pm 0.04$ | $3.65 \pm 0.04$ | $3.24 \pm 0.03$ | $3.64 \pm 0.02$ | $3.11 \pm 0.03$ | $3.49 \pm 0.04$ |
| | $1000 \to 3000$ | $4.22 \pm 0.07$ | $4.26 \pm 0.06$ | $4.05 \pm 0.09$ | $4.18 \pm 0.05$ | $4.00 \pm 0.05$ | $4.12 \pm 0.04$ | $4.14 \pm 0.05$ | $4.20 \pm 0.03$ | $3.96 \pm 0.04$ | $4.04 \pm 0.03$ |
| USHCN (m) | $24 \to 1$ | $5.06 \pm 0.02$ | $3.06 \pm 0.09$ | $4.94 \pm 0.03$ | $3.02 \pm 0.10$ | $5.00 \pm 0.04$ | $3.04 \pm 0.08$ | $5.01 \pm 0.04$ | $3.04 \pm 0.11$ | $4.84 \pm 0.02$ | $2.92 \pm 0.06$ |
| | $24 \to 6$ | $4.99 \pm 0.05$ | $3.01 \pm 0.07$ | $5.05 \pm 0.01$ | $3.08 \pm 0.06$ | $4.98 \pm 0.05$ | $3.10 \pm 0.04$ | $4.84 \pm 0.04$ | $2.95 \pm 0.04$ | $4.81 \pm 0.04$ | $2.94 \pm 0.04$ |
| | $24 \to 12$ | $5.09 \pm 0.02$ | $3.06 \pm 0.08$ | $4.98 \pm 0.08$ | $2.97 \pm 0.07$ | $5.05 \pm 0.05$ | $3.08 \pm 0.04$ | $4.96 \pm 0.03$ | $3.04 \pm 0.06$ | $4.81 \pm 0.05$ | $2.93 \pm 0.03$ |
| PhysioNet (h) | $12 \to 36$ | $6.44 \pm 0.14$ | $4.23 \pm 0.05$ | $6.71 \pm 0.21$ | $4.39 \pm 0.09$ | $6.12 \pm 0.08$ | $4.03 \pm 0.06$ | $6.46 \pm 0.17$ | $4.23 \pm 0.08$ | $6.08 \pm 0.14$ | $3.99 \pm 0.08$ |
| | $24 \to 24$ | $5.23 \pm 0.15$ | $3.78 \pm 0.06$ | $5.15 \pm 0.09$ | $3.63 \pm 0.11$ | $4.82 \pm 0.13$ | $3.55 \pm 0.09$ | $4.96 \pm 0.16$ | $3.72 \pm 0.12$ | $4.99 \pm 0.18$ | $3.62 \pm 0.07$ |
| | $36 \to 12$ | $4.36 \pm 0.27$ | $3.58 \pm 0.04$ | $4.44 \pm 0.30$ | $3.67 \pm 0.04$ | $3.95 \pm 0.18$ | $3.23 \pm 0.04$ | $4.37 \pm 0.11$ | $3.72 \pm 0.12$ | $3.81 \pm 0.12$ | $3.18 \pm 0.07$ |
| MIMIC-III (h) | $12 \to 36$ | $1.85 \pm 0.02$ | $7.71 \pm 0.13$ | $1.91 \pm 0.01$ | $7.80 \pm 0.05$ | $1.86 \pm 0.04$ | $7.69 \pm 0.07$ | $1.86 \pm 0.04$ | $7.74 \pm 0.07$ | $1.80 \pm 0.02$ | $7.54 \pm 0.06$ |
| | $24 \to 24$ | $1.71 \pm 0.04$ | $7.19 \pm 0.12$ | $1.71 \pm 0.02$ | $7.07 \pm 0.03$ | $1.66 \pm 0.02$ | $6.94 \pm 0.01$ | $1.66 \pm 0.03$ | $7.19 \pm 0.04$ | $1.63 \pm 0.04$ | $6.83 \pm 0.08$ |
| | $36 \to 12$ | $1.58 \pm 0.03$ | $7.03 \pm 0.18$ | $1.64 \pm 0.02$ | $6.98 \pm 0.13$ | $1.60 \pm 0.04$ | $6.89 \pm 0.19$ | $1.54 \pm 0.05$ | $6.75 \pm 0.07$ | $1.48 \pm 0.02$ | $6.56 \pm 0.06$ |

*Table D.3.3.* Forecasting performance of embedding variants with QuITE$^{++}$.

# E  Computational Complexity Analysis

## E.1  Theoretical Analysis

**Notation.**  We analyze the computational complexity of the proposed irregular time series embedding method and compare it with conventional MLP-based input embedding modules. Let $B$ denote the batch size, $N$ the number of variables, $M$ the number of temporal patches, and $D$ the embedding dimension. Since the sequence length has different meanings in variable-based and patch-based embeddings, we distinguish them as follows.

- $L_v$: the sequence length in variable-based embedding, i.e., the number of irregular observations for each variable;

- $L_p$: the sequence length within each patch in patch-based embedding, i.e., the number of observation tokens contained in a single patch.

In general, $L_p \ll L_v$, and $L_p$ is a small constant determined by the model design.

**Conventional Input Embedding.**  For conventional input embedding, each observation is independently projected into a $D$-dimensional latent space using a linear transformation.

*Variable-based conventional embedding.* In the variable-based setting, each variable contains $L_v$ observation tokens. The overall computational complexity therefore scales linearly with the sequence length as

$$\mathcal{O}(BNL_vD).$$

*Patch-based conventional embedding.* In the patch-based setting, the same MLP embedding is applied independently to each patch. With $M$ patches and $L_p$ tokens per patch, the total computational complexity becomes

$$\mathcal{O}(BMNL_pD).$$

**Proposed Query-Based Embedding.**  Each observation is first represented as the sum of value and time embeddings. This tokenization stage incurs a linear cost of $\mathcal{O}(BNL_vD)$ or $\mathcal{O}(BMNL_pD)$ for variable-based and patch-based embeddings, respectively. Since this cost is dominated by the subsequent attention operation, it is omitted from the dominant-term analysis.

*Variable-based query embedding.* In the proposed variable-based formulation, one learnable variable token is appended to the $L_v$ observation tokens, yielding a sequence of length $L_v + 1$. A single Transformer block is then applied to perform attention-based aggregation. The resulting computational complexity is

$$\mathcal{O}\big(BN\big((L_v + 1)^2D + (L_v + 1)D^2\big)\big).$$

*Patch-based query embedding.* Similarly, in the patch-based formulation, a learnable patch token is introduced for each patch–variable pair. Attention is applied to sequences of length $L_p + 1$, resulting in an overall complexity of

$$\mathcal{O}\big(BMN\big((L_p + 1)^2D + (L_p + 1)D^2\big)\big).$$

Although the proposed embedding introduces quadratic terms with respect to $L_v$ or $L_p$ due to the self-attention mechanism, the additional overhead remains limited in practice. The attention module is applied only once at the input embedding stage, and in the patch-based setting the patch length $L_p$ is typically a small fixed constant. In return, the proposed approach enables mask-aware aggregation of irregular and asynchronous observations without interpolation, providing substantially richer representations while preserving compatibility with standard MTS backbones.

## E.2 Empirical Analysis

| Dataset | Horizon | Model | MSE | | FLOPs | | Parameters | | Training Time | | Inference Time | |
|---|---|---|---|---|---|---|---|---|---|---|---|---|
| | | | Base | +QuITE | Base | +QuITE | Base | +QuITE | Base | +QuITE | Base | +QuITE |
| Human Activity | 1000ms → 3000ms | PatchTST | 5.06 | 4.69 | 16.5G | 1.16G | 1718273 | 126657 | 0.97 | 0.81 | 0.17 | 0.14 |
| | | PatchMixer | 4.80 | 4.71 | 14.6G | 1.05G | 530761 | 50953 | 0.77 | 0.59 | 0.14 | 0.12 |
| | | TMix | 4.97 | 4.75 | 4.12G | 1.14G | 1217665 | 322305 | 0.59 | 0.61 | 0.12 | 0.12 |
| | | iTransformer | 5.71 | 4.10 | 15.8G | 1.04G | 4048641 | 127169 | 0.73 | 0.79 | 0.11 | 0.14 |
| | | S-Mamba | 5.37 | 4.20 | 15.9G | 1.06G | 4792897 | 187905 | 4.10 | 12.70 | 0.68 | 2.12 |
| | | TimeXer | 4.77 | 4.04 | 17.3G | 1.29G | 2521857 | 194497 | 1.04 | 1.10 | 0.18 | 0.17 |
| | | QuITE++ | - | 3.96 | - | 2.49G | - | 214721 | - | 1.62 | - | 0.19 |
| | | Hi-Patch | 4.20 | - | 473M | - | 363137 | - | 9.84 | - | 1.56 | - |
| | | HyperIMTS | 4.00 | - | 38.2G | - | 809090 | - | 7.07 | - | 0.86 | - |
| | 2000ms → 2000ms | PatchTST | 4.02 | 3.62 | 11.9G | 911M | 1720321 | 126657 | 1.27 | 1.16 | 0.23 | 0.20 |
| | | PatchMixer | 3.76 | 3.67 | 9.94G | 797M | 532809 | 50953 | 1.00 | 0.88 | 0.19 | 0.18 |
| | | TMix | 3.87 | 3.66 | 2.94G | 886M | 1218689 | 322305 | 0.81 | 0.89 | 0.17 | 0.18 |
| | | iTransformer | 4.88 | 3.25 | 11.3G | 786M | 4048641 | 127169 | 0.91 | 1.18 | 0.15 | 0.20 |
| | | S-Mamba | 4.68 | 3.37 | 11.3G | 792M | 4792897 | 187905 | 5.09 | 10.21 | 1.52 | 3.37 |
| | | TimeXer | 3.87 | 3.19 | 12.6G | 1.10G | 2532097 | 194497 | 1.48 | 1.62 | 0.25 | 0.25 |
| | | QuITE++ | - | 3.11 | - | 1.87G | - | 55137 | - | 1.94 | - | 0.26 |
| | | Hi-Patch | 3.26 | - | 337M | - | 363137 | - | 5.42 | - | 0.64 | - |
| | | HyperIMTS | 3.15 | - | 29.2G | - | 809090 | - | 3.89 | - | 1.00 | - |
| | 3000ms → 1000ms | PatchTST | 3.10 | 2.76 | 7.05G | 653M | 1722369 | 126657 | 2.31 | 2.33 | 0.43 | 0.40 |
| | | PatchMixer | 2.84 | 2.78 | 5.23G | 539M | 534857 | 50953 | 1.66 | 1.72 | 0.35 | 0.35 |
| | | TMix | 2.93 | 2.77 | 1.72G | 638M | 1219713 | 322305 | 1.60 | 1.76 | 0.34 | 0.35 |
| | | iTransformer | 3.77 | 2.58 | 6.41G | 563M | 4048641 | 127169 | 1.63 | 2.28 | 0.25 | 0.40 |
| | | S-Mamba | 3.56 | 2.72 | 6.64G | 544M | 4792897 | 187905 | 3.19 | 6.17 | 1.53 | 2.53 |
| | | TimeXer | 2.99 | 2.53 | 7.71G | 891M | 2542337 | 194497 | 2.82 | 3.18 | 0.47 | 0.48 |
| | | QuITE++ | - | 2.46 | - | 1.15G | - | 55137 | - | 3.76 | - | 0.55 |
| | | Hi-Patch | 2.56 | - | 196M | - | 363137 | - | 7.05 | - | 3.67 | - |
| | | HyperIMTS | 2.49 | - | 21.0G | - | 809090 | - | 5.21 | - | 1.51 | - |
| USHCN | 24m → 1m | PatchTST | 5.35 | 5.07 | 16.6G | 2.94G | 1721857 | 127425 | 6.83 | 7.09 | 1.73 | 1.74 |
| | | PatchMixer | 5.22 | 5.11 | 4.52G | 2.03G | 534753 | 52129 | 5.31 | 6.49 | 1.56 | 1.64 |
| | | TMix | 5.41 | 5.18 | 12.1G | 4.68G | 16954241 | 4258305 | 5.34 | 6.59 | 1.50 | 1.67 |
| | | iTransformer | 5.91 | 5.06 | 4.70G | 1.80G | 1810945 | 126721 | 2.97 | 7.73 | 0.67 | 1.79 |
| | | S-Mamba | 5.73 | 5.04 | 4.06G | 1.83G | 2555201 | 194497 | 11.14 | 15.13 | 2.02 | 3.86 |
| | | TimeXer | 5.36 | 4.97 | 18.0G | 4.77G | 2608641 | 194817 | 7.22 | 10.12 | 1.78 | 2.05 |
| | | QuITE++ | - | 4.84 | - | 682M | - | 42465 | - | 8.61 | - | 1.80 |
| | | Hi-Patch | 5.00 | - | 281M | - | 400001 | - | 22.40 | - | 5.13 | - |
| | | HyperIMTS | 4.96 | - | 49.6G | - | 429698 | - | 13.88 | - | 4.40 | - |
| | 24m → 6m | PatchTST | 5.30 | 5.06 | 27.5G | 3.52G | 1721857 | 127425 | 1.18 | 1.08 | 0.28 | 0.26 |
| | | PatchMixer | 5.31 | 5.02 | 16.4G | 2.69G | 534753 | 52129 | 0.91 | 0.99 | 0.24 | 0.25 |
| | | TMix | 7.31 | 5.52 | 14.7G | 5.35G | 16954241 | 4258305 | 0.86 | 1.02 | 0.23 | 0.26 |
| | | iTransformer | 5.51 | 4.86 | 16.1G | 2.67G | 1810945 | 126721 | 0.59 | 1.18 | 0.13 | 0.27 |
| | | S-Mamba | 5.50 | 4.93 | 15.7G | 2.55G | 2555201 | 194497 | 2.67 | 7.07 | 0.44 | 2.51 |
| | | TimeXer | 5.21 | 4.98 | 30.2G | 5.18G | 2608641 | 194817 | 1.26 | 1.57 | 0.29 | 0.32 |
| | | QuITE++ | - | 4.81 | - | 1.12G | - | 55297 | - | 1.34 | - | 0.28 |
| | | Hi-Patch | 5.13 | - | 615M | - | 400001 | - | 7.69 | - | 1.17 | - |
| | | HyperIMTS | 4.99 | - | 82.1G | - | 429698 | - | 4.91 | - | 0.85 | - |
| | 24m → 12m | PatchTST | 5.11 | 5.04 | 38.9G | 4.26G | 1721857 | 127425 | 0.68 | 0.57 | 0.15 | 0.13 |
| | | PatchMixer | 5.22 | 5.00 | 28.1G | 3.45G | 534753 | 52129 | 0.54 | 0.53 | 0.13 | 0.13 |
| | | TMix | 7.53 | 5.03 | 17.5G | 6.28G | 16954241 | 4258305 | 0.47 | 0.53 | 0.12 | 0.13 |
| | | iTransformer | 5.46 | 4.94 | 26.5G | 3.24G | 1810945 | 126721 | 0.37 | 0.60 | 0.08 | 0.14 |
| | | S-Mamba | 5.40 | 4.93 | 27.6G | 3.31G | 2555201 | 194497 | 3.08 | 9.72 | 0.51 | 1.62 |
| | | TimeXer | 5.23 | 4.93 | 40.3G | 6.25G | 2608641 | 194817 | 0.71 | 0.80 | 0.15 | 0.16 |
| | | QuITE++ | - | 4.81 | - | 6.25G | - | 29633 | - | 0.66 | - | 0.14 |
| | | Hi-Patch | 5.04 | - | 957M | - | 400001 | - | 3.77 | - | 0.61 | - |
| | | HyperIMTS | 4.97 | - | 123G | - | 429698 | - | 3.22 | - | 0.50 | - |

*Table E.1.* Empirical Computational Analysis on Human Activity and USHCN.

| Dataset | Horizon | Model | MSE | | FLOPs | | Parameters | | Training Time | | Inference Time | |
|---|---|---|---|---|---|---|---|---|---|---|---|---|
| | | | Base | +QuITE | Base | +QuITE | Base | +QuITE | Base | +QuITE | Base | +QuITE |
| PhysioNet | 12h → 36h | PatchTST | 20.03 | 17.47 | 91.1G | 8.97G | 1729537 | 126529 | 14.06 | 8.38 | 2.20 | 1.53 |
| | | PatchMixer | 18.62 | 17.52 | 99.4G | 10.6G | 541985 | 50785 | 11.80 | 7.84 | 1.85 | 1.50 |
| | | TMix | 18.57 | 17.48 | 27.3G | 9.32G | 435841 | 125057 | 7.07 | 7.63 | 1.36 | 1.48 |
| | | iTransformer | 21.14 | 6.32 | 133G | 13.0G | 1771521 | 129025 | 10.02 | 7.45 | 1.41 | 1.47 |
| | | S-Mamba | 8.23 | 6.26 | 111G | 9.31G | 2515777 | 189761 | 29.47 | 26.61 | 4.91 | 4.44 |
| | | TimeXer | 6.91 | 6.18 | 141G | 13.3G | 2558465 | 196225 | 13.79 | 10.74 | 2.10 | 1.86 |
| | | QuITE++ | - | 6.08 | - | 24.5G | - | 216577 | - | 12.18 | - | 2.09 |
| | | Hi-Patch | 6.39 | - | 3.71G | - | 102657 | - | 19.13 | - | 4.57 | - |
| | | HyperIMTS | 6.11 | - | 19.6G | - | 812802 | - | 3.22 | - | 0.50 | - |
| | 24h → 24h | PatchTST | 15.28 | 10.62 | 75.2G | 10.6G | 1729537 | 126657 | 12.03 | 10.08 | 1.97 | 1.82 |
| | | PatchMixer | 12.28 | 11.88 | 65.9G | 8.33G | 542025 | 50953 | 9.45 | 9.28 | 1.63 | 1.73 |
| | | TMix | 12.38 | 10.72 | 16.2G | 12.7G | 1223297 | 322305 | 6.10 | 9.28 | 1.29 | 1.74 |
| | | iTransformer | 16.48 | 4.99 | 107G | 13.8G | 1771521 | 129025 | 7.56 | 10.51 | 1.09 | 1.89 |
| | | S-Mamba | 6.93 | 5.11 | 63.2G | 8.54G | 2515777 | 189761 | 17.17 | 24.81 | 2.86 | 4.14 |
| | | TimeXer | 5.79 | 4.91 | 88.1G | 17.2G | 2585601 | 196353 | 11.38 | 16.11 | 1.85 | 2.60 |
| | | QuITE++ | - | 4.99 | - | 18.1G | - | 216577 | - | 19.58 | - | 2.93 |
| | | Hi-Patch | 5.07 | - | 3.61G | - | 140097 | - | 31.73 | - | 7.54 | - |
| | | HyperIMTS | 4.65 | - | 13.5G | - | 812802 | - | 5.34 | - | 0.82 | - |
| | 36h → 12h | PatchTST | 12.97 | 8.87 | 46.8G | 9.88G | 1734401 | 126657 | 9.44 | 10.45 | 1.70 | 1.93 |
| | | PatchMixer | 10.33 | 9.06 | 42.1G | 6.98G | 546889 | 50953 | 6.98 | 9.70 | 1.40 | 1.84 |
| | | TMix | 10.50 | 8.96 | 12.8G | 12.9G | 1225729 | 322305 | 4.89 | 9.59 | 1.15 | 1.82 |
| | | iTransformer | 14.09 | 4.33 | 56.5G | 11.7G | 1771521 | 129025 | 4.86 | 12.78 | 0.77 | 2.16 |
| | | S-Mamba | 6.26 | 4.11 | 38.3G | 7.95G | 2515777 | 189761 | 7.75 | 34.29 | 1.29 | 5.71 |
| | | TimeXer | 4.94 | 4.06 | 51.9G | 17.4G | 2609921 | 196353 | 8.99 | 19.22 | 1.63 | 3.05 |
| | | QuITE++ | - | 3.81 | - | 16.9G | - | 216449 | - | 12.18 | - | 2.20 |
| | | Hi-Patch | 4.27 | - | 2.26G | - | 140097 | - | 32.46 | - | 8.87 | - |
| | | HyperIMTS | 3.99 | - | 7.91G | - | 812802 | - | 13.52 | - | 4.35 | - |
| MIMIC-III | 12h → 36h | PatchTST | 5.67 | 5.37 | 163G | 5.30G | 1737729 | 126529 | 96.95 | 40.71 | 12.66 | 6.11 |
| | | PatchMixer | 5.53 | 5.37 | 200G | 16.9G | 550177 | 50785 | 89.51 | 37.41 | 12.22 | 5.60 |
| | | TMix | 5.57 | 5.41 | 30.0G | 9.13G | 439937 | 125057 | 47.25 | 36.19 | 6.85 | 5.60 |
| | | iTransformer | 6.00 | 1.83 | 81.3G | 5.13G | 1880833 | 132545 | 83.87 | 34.27 | 11.13 | 5.67 |
| | | S-Mamba | 2.83 | 1.82 | 87.0G | 5.42G | 2625089 | 193281 | 107.35 | 199.51 | 17.89 | 33.25 |
| | | TimeXer | 2.01 | 1.84 | 109G | 21.0G | 2597121 | 199745 | 243.47 | 47.16 | 31.76 | 6.87 |
| | | QuITE++ | - | 1.80 | - | 6.20G | - | 57761 | - | 71.74 | - | 10.76 |
| | | Hi-Patch | 1.88 | - | 7.37G | - | 408961 | - | 131.29 | - | 29.18 | - |
| | | HyperIMTS | 1.85 | - | 28.6G | - | 812802 | - | 83.90 | - | 9.80 | - |
| | 24h → 24h | PatchTST | 4.33 | 3.90 | 59.2G | 2.05G | 1768193 | 126529 | 45.91 | 39.36 | 7.07 | 6.16 |
| | | PatchMixer | 4.10 | 3.94 | 73.2G | 2.73G | 552113 | 50785 | 31.82 | 33.52 | 5.50 | 5.46 |
| | | TMix | 4.14 | 3.98 | 21.0G | 1.27G | 455169 | 125057 | 32.14 | 32.87 | 5.41 | 5.47 |
| | | iTransformer | 5.50 | 1.67 | 23.6G | 2.25G | 1880833 | 132545 | 52.81 | 37.44 | 7.33 | 6.09 |
| | | S-Mamba | 2.38 | 1.64 | 27.7G | 2.94G | 2625089 | 193281 | 94.17 | 175.55 | 15.70 | 39.26 |
| | | TimeXer | 1.92 | 1.68 | 6.23G | 2.36G | 2688513 | 199745 | 328.59 | 56.44 | 41.75 | 8.08 |
| | | QuITE++ | - | 1.63 | - | 2.89G | - | 57761 | - | 113.04 | - | 13.74 |
| | | Hi-Patch | 1.70 | - | 1.83G | - | 408961 | - | 355.54 | - | 103.01 | - |
| | | HyperIMTS | 1.68 | - | 15.9G | - | 812802 | - | 158.70 | - | 23.32 | - |
| | 36h → 12h | PatchTST | 4.87 | 3.82 | 34.9G | 5.70G | 1739521 | 126913 | 66.11 | 43.22 | 9.53 | 6.71 |
| | | PatchMixer | 4.11 | 3.87 | 10.4G | 705M | 580641 | 51313 | 54.70 | 35.95 | 8.09 | 6.06 |
| | | TMix | 4.04 | 3.87 | 1.86G | 1.98G | 4376065 | 1110017 | 24.16 | 36.62 | 4.60 | 6.04 |
| | | iTransformer | 6.05 | 1.56 | 11.2G | 1.87G | 1880833 | 132545 | 29.88 | 50.86 | 4.22 | 7.51 |
| | | S-Mamba | 3.92 | 1.52 | 8.25G | 2.43G | 2625089 | 193281 | 52.59 | 148.01 | 8.77 | 24.67 |
| | | TimeXer | 1.98 | 1.55 | 21.6G | 6.16G | 2743809 | 200129 | 312.59 | 78.64 | 40.80 | 10.66 |
| | | QuITE++ | - | 1.48 | - | 549M | - | 32289 | - | 129.17 | - | 16.22 |
| | | Hi-Patch | 1.56 | - | 948M | - | 706177 | - | 319.38 | - | 83.91 | - |
| | | HyperIMTS | 1.52 | - | 5.52G | - | 812802 | - | 142.55 | - | 28.19 | - |

*Table E.2.* Empirical Computational Analysis on PhysioNet and MIMIC-III.

# F Visualization

## F.1 Embedding Visualizations With and Without QuITE

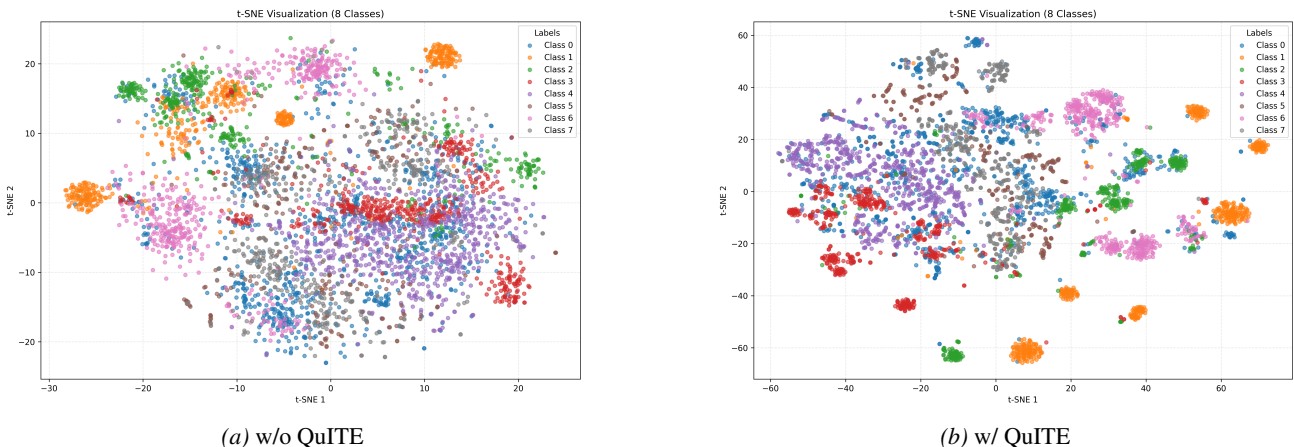

*(a)* w/o QuITE                                          *(b)* w/ QuITE

*Figure F.1.1.* Visualization of PatchTST Embedding Representations

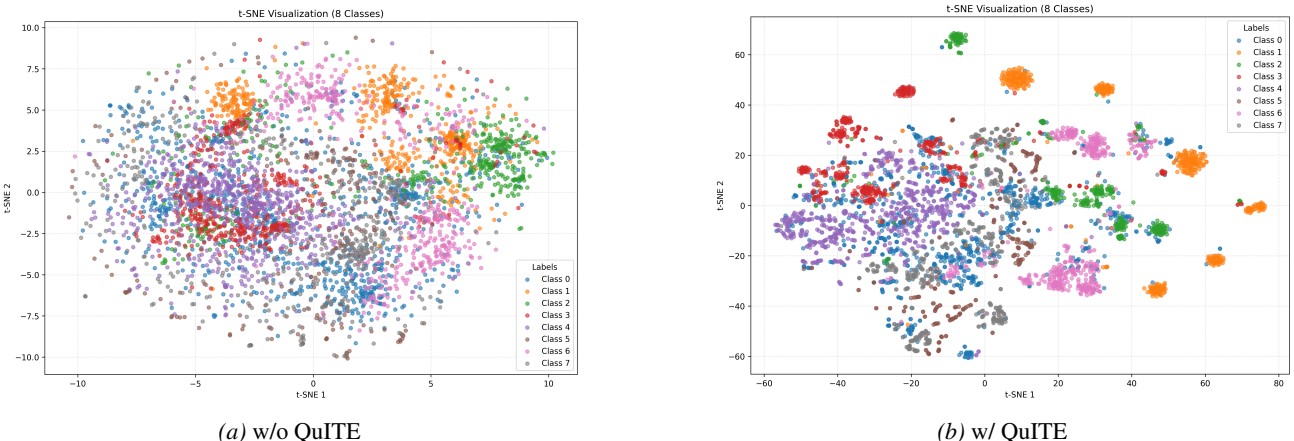

*(a)* w/o QuITE                                          *(b)* w/ QuITE

*Figure F.1.2.* Visualization of PatchMixer Embedding Representations

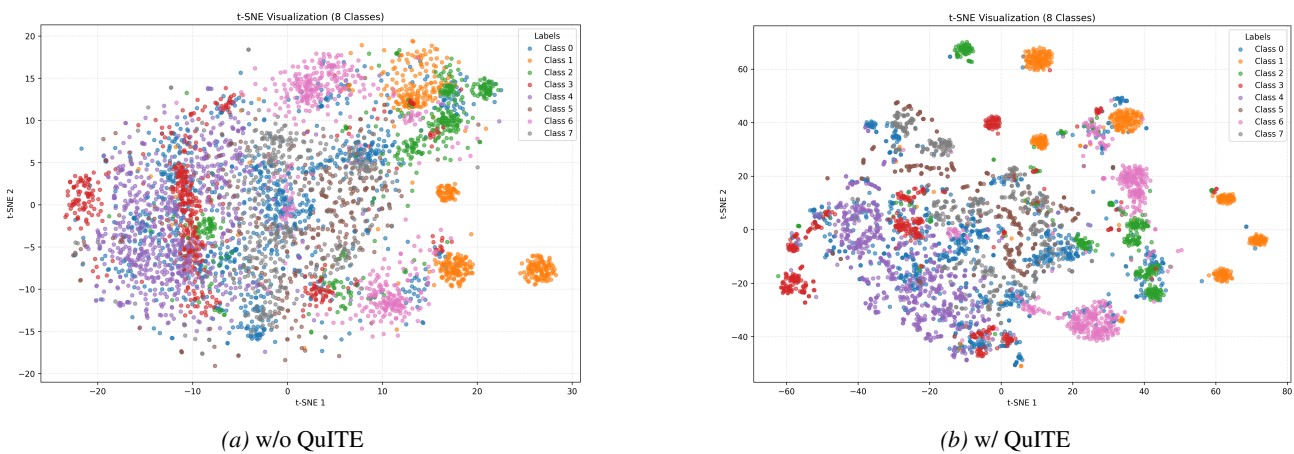

*(a)* w/o QuITE                                          *(b)* w/ QuITE

*Figure F.1.3.* Visualization of TMix Embedding Representations

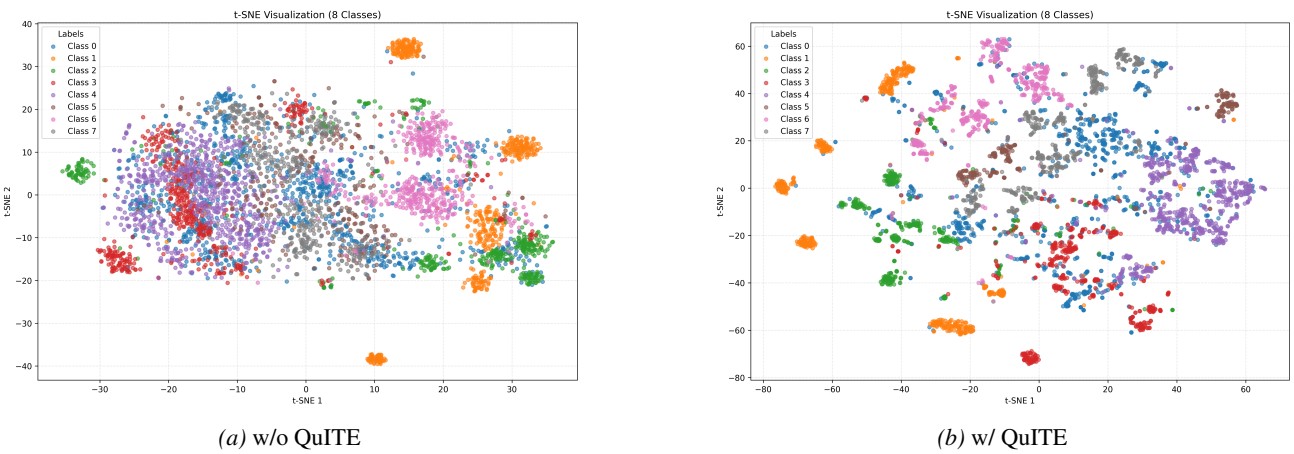

*(a)* w/o QuITE    *(b)* w/ QuITE

*Figure F.1.4.* Visualization of iTransformer Embedding Representations

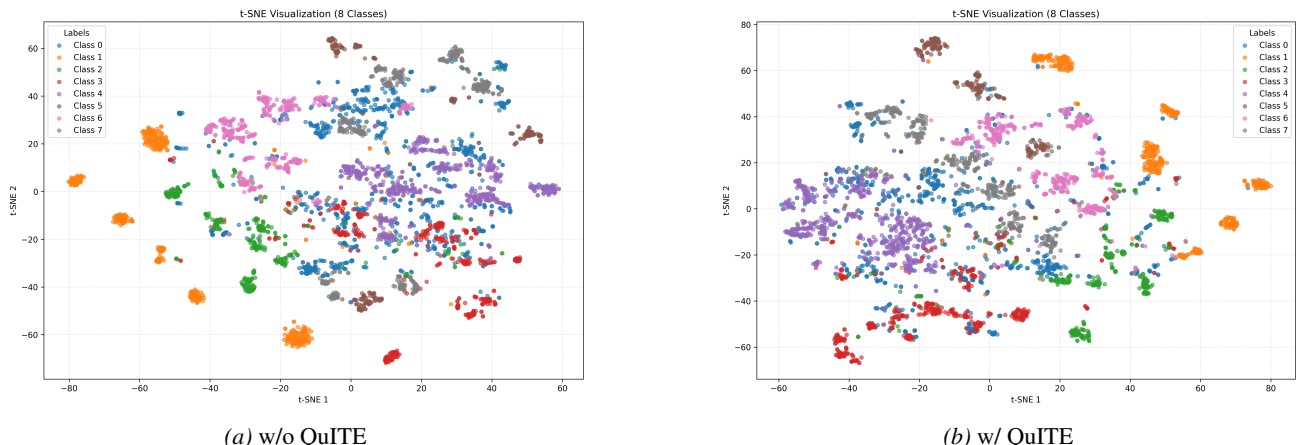

*(a)* w/o QuITE    *(b)* w/ QuITE

*Figure F.1.5.* Visualization of S-Mamba Embedding Representations

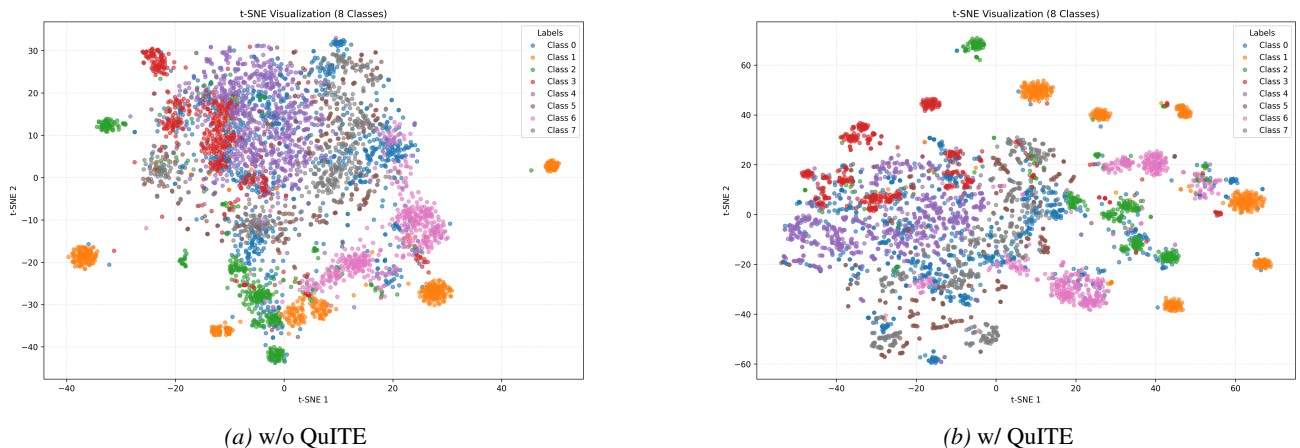

*(a)* w/o QuITE    *(b)* w/ QuITE

*Figure F.1.6.* Visualization of TimeXer's Patch Embedding Representations

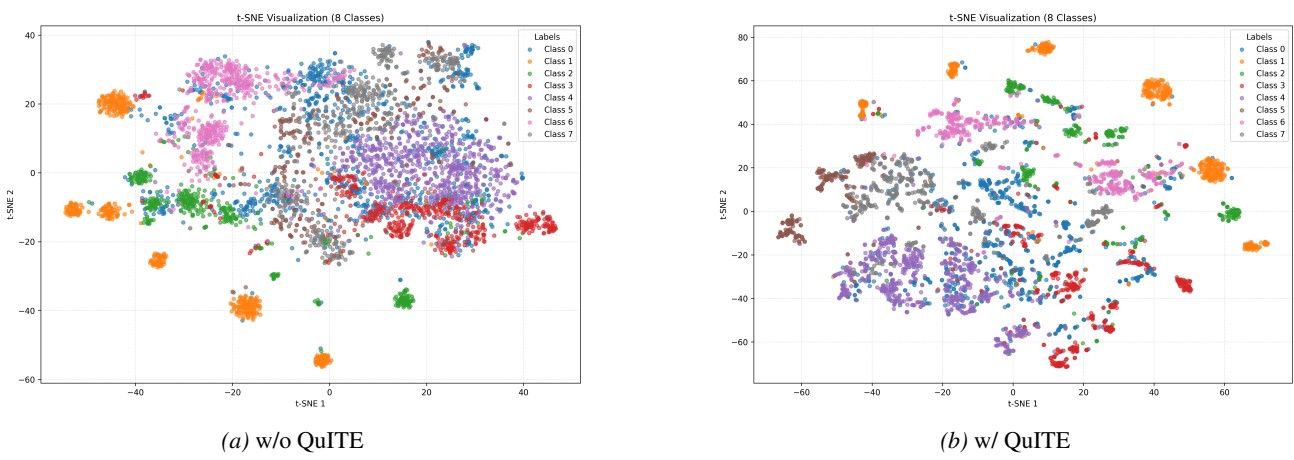

*(a)* w/o QuITE                                      *(b)* w/ QuITE

*Figure F.1.7.* Visualization of TimeXer's Variable Embedding Representations

## F.2 Forecasting Visualization With and Without QuITE

We further provide qualitative forecasting comparisons by applying QuITE to representative patch-based, variable-based, and hybrid backbones, namely PatchTST, iTransformer, and TimeXer. Across diverse datasets, QuITE-equipped models produce predictions that more closely follow the ground-truth trajectories. These qualitative results further support the effectiveness of QuITE.

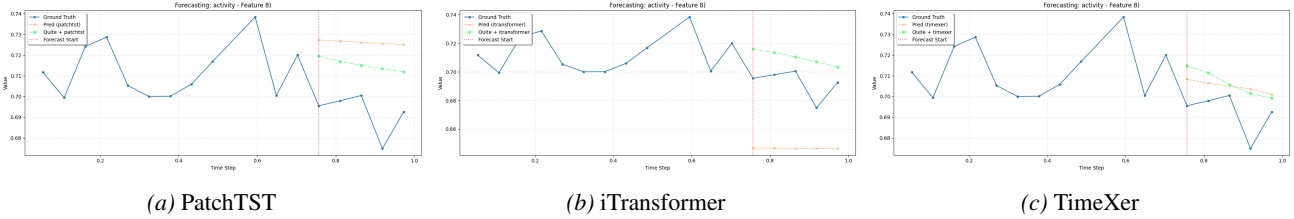

| *(a)* PatchTST | *(b)* iTransformer | *(c)* TimeXer |

*Figure F.2.1.* Human Activity Forecasting Visualization(3000ms → 1000ms)

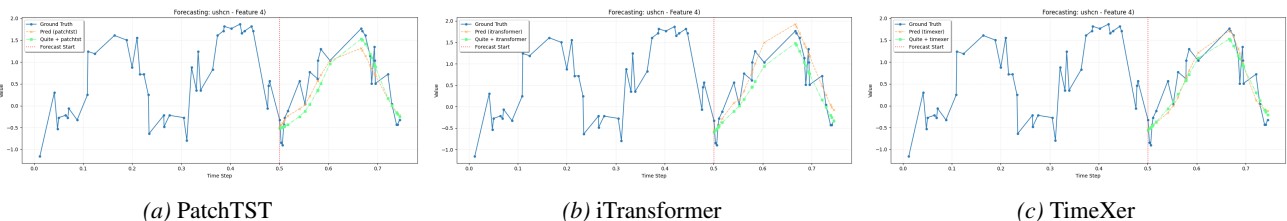

| *(a)* PatchTST | *(b)* iTransformer | *(c)* TimeXer |

*Figure F.2.2.* USHCN Forecasting Visualization (24months → 12months)

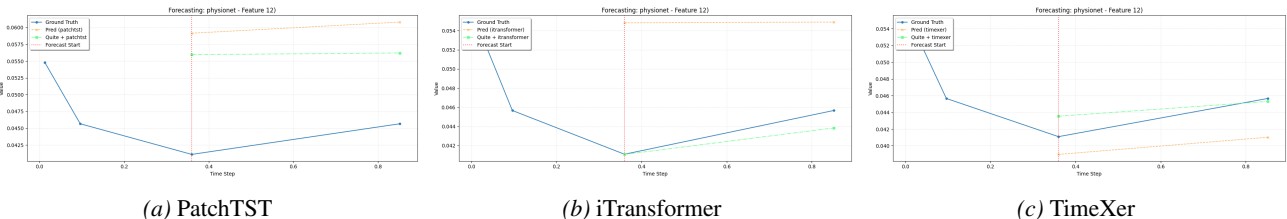

| *(a)* PatchTST | *(b)* iTransformer | *(c)* TimeXer |

*Figure F.2.3.* PhysioNet Forecasting Visualization (12h → 36h)

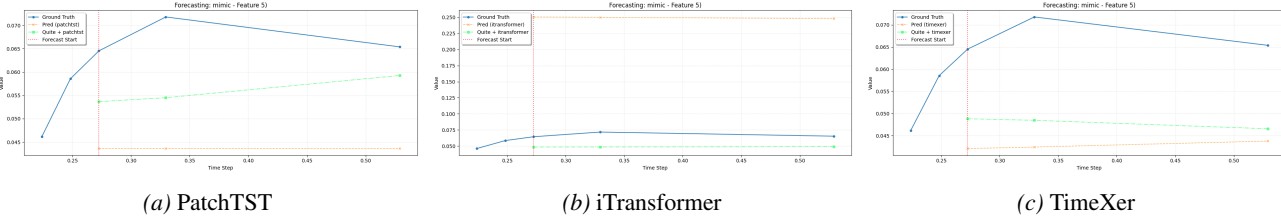

| *(a)* PatchTST | *(b)* iTransformer | *(c)* TimeXer |

*Figure F.2.4.* MIMIC-III Forecasting Visualization (12h → 36h)

