# OpenReview forum: "QuITE: Query-Based Irregular Time Series Embedding"
_ICML.cc/2026/Conference — ICML 2026 regular_

### Official Review · Reviewer_QALz · 2026-03-11

**Soundness:** 3
**Presentation:** 2
**Significance:** 3
**Originality:** 3
**Overall Recommendation:** 4
**Confidence:** 5

**Summary:**

This paper addresses the challenge of modeling Irregular Multivariate Time Series (IMTS). The authors propose QuITE, a plug-and-play, backbone-agnostic input embedding module that utilizes learnable query tokens and self-attention to aggregate irregular observations. This approach translates irregular data into fixed-dimensional representations without introducing artificial values, enable time series models to process IMTS directly. Extensive experiments across multiple benchmarks and architectures demonstrate that QuITE consistently and significantly improves forecasting performance.

**Compliance With Llm Reviewing Policy:**

Affirmed.

**Final Justification:**

The rebuttal addressed all my concerns. The paper originally overclaimed their efficiency, but the author provides supported evidence to correct their statement. Multiple experiments have demonstrated that the proposed module is a plug-and-play module for most transformer-based models. Most improvement is consistent, and it can be transferred to irregular time series classification problem.

**Key Questions For Authors:**

The learnable token works as a summary of all time series token, but typical summary operation is usually lossy. How can a summarized representation improve the performance significantly over the baseline?

**Limitations:**

No limitation is discussed. the effectiveness of proposed method is not clear for classification task, and the method usually increase training/inference complexity over the backbone model.

**Strengths And Weaknesses:**

Strength

1. QuITE operates purely at the input embedding level, allowing practitioners to seamlessly integrate it with diverse, state-of-the-art MTS backbones without needing to modify the core model architectures.

2. The module demonstrates consistent performance gains across four real-world datasets to show the effectivenss of proposed method.


Weakness

1. Irregular time series classificaiton is also an important task, while authors claimed a strong irregular time series embedding without implementing any classification task. All dataset used in the paper can be used for classifcation. Without such evaluation, it will degrade the scope of the paper to irregular time series forecasting.

2. QuITE indeed improves over the tranformer based backbone and the authors claim that QuITE is a plug-and-play module, but there is QuITE++ model archives the SOTA performance with specific architecture design. I did not see any ablation on QuITE++ encoder. It is difficult to tell if the designed encoder or QuITE contribute most performance to achive SOTA

3. QuITE introduces more training/inference time, but there is no specific complexity analysis (FLOPs, number of parameter increased)

4. The learnable token works as a summary of all time series token, but typical summary operation is usually lossy. How can a summarized representation improve the performance significantly over the baseline?

5. To prove the applicability, I would like to see that if controled the complexity of model. For example, Complexity(PatchTST + QuITE) = Complexity(PatchTST). Given the fact that QuITE usually increase complexity, we may reduce the hidden dimension of backbone to make equality valid. The QuITE has a certain potential if we can still see improved performance under the same complexity

---

> ### Author Rebuttal · Authors · 2026-03-31
>
> **W1 / Q1. Scope beyond forecasting / classification tasks.**
>
> Thank you for this important comment. We agree that evaluating only forecasting may make the scope appear narrower than intended. To address this, we additionally evaluated QuITE on three classification benchmarks (P12, P19, PAM). QuITE consistently improves the corresponding backbones, with mean relative gains of about 5.2%–15.8% across backbone families. This suggests that, although our main experiments focus on forecasting, QuITE also generalizes well to classification. Due to space limitations, we report representative classification results here. In the revision, we will include a more comprehensive summary (e.g., PAM results and overall averages).
>
> | DataSet | Metric   | PatchTST | + QuITE | Imp.   | PatchMixer | + QuITE | Imp.    | TMix  | + QuITE | Imp.    |
> | ------- | -------- | -------- | ------- | ------ | ---------- | ------- | ------- | ----- | ------- | ------- |
> | P12     | AUROC    | 83.6     | 84.5    | 1.10%  | 78.2       | 83.9    | 7.30%   | 82.8  | 85.2    | 2.90%   |
> |         | AUPR     | 47       | 54.5    | 16.00% | 39.2       | 45.2    | 15.30%  | 45.5  | 51.9    | 14.10%  |
> | P19     | AUROC    | 82.4     | 85      | 3.20%  | 75         | 83.8    | 11.70%  | 63    | 81.2    | 28.90%  |
> |         | AUPR     | 40.3     | 54.5    | 35.20% | 26.4       | 55.8    | 111.40% | 10.1  | 38      | 276.20% |
> | PAM     | F1 | 86.3     | 89.3    | 3.50%  | 75.7       | 83.7    | 10.60%  | 81.4  | 85.9    | 5.50%   |
>
> | DataSet | Metric   | iTransformer | + QuITE | Imp.   | S-Mamba | + QuITE | Imp.   | TimeXer | + QuITE | Imp.   |
> | ------- | -------- | ------------ | ------- | ------ | ------- | ------- | ------ | ------- | ------- | ------ |
> | P12     | AUROC    | 82.6         | 85.3    | 3.30%  | 82.3    | 84.1    | 2.20%  | 83.5    | 86.1    | 3.10%  |
> |         | AUPR     | 45.3         | 47.6    | 5.10%  | 43.6    | 49.5    | 13.50% | 48      | 49.5    | 3.10%  |
> | P19     | AUROC    | 82.5         | 86.5    | 4.80%  | 80.2    | 85.4    | 6.50%  | 83.2    | 85.3    | 2.50%  |
> |         | AUPR     | 39.2         | 51.7    | 31.90% | 37      | 51.7    | 39.70% | 41.9    | 54.9    | 31.00% |
> | PAM     | F1 | 89.2         | 91.5    | 2.60%  | 85.9    | 88.7    | 3.30%  | 87      | 90.3    | 3.80%  |
>
> ---
> **W2. Contribution of QuITE vs. QuITE++ encoder.**
>
> We agree that this point should be clarified more explicitly. QuITE++ remains a **Transformer-style MTS backbone** and is structurally close to **TimeXer**, with its main change being the replacement of the second cross-variable cross-attention with self-attention. Thus, it is not a fundamentally separate modeling paradigm.
>
> More importantly, **Table 5 and Appendix E isolate the contribution of the embedding module**: under the same QuITE++ encoder, the **QuITE-based variant** outperforms the others. This indicates that the final SOTA performance is driven primarily by the **QuITE embedding**, rather than by the encoder alone.
>
> ---
> **W3 / Q1 / W5. Complexity analysis and controlled applicability.**
>
> We agree that this point deserves clearer discussion. The submission already includes a dedicated complexity analysis in Appendix F.
> - Theoretical: Although QuITE introduces quadratic terms through self-attention, the practical overhead remains limited because attention is applied **only once at the input embedding stage**.
> - Empirical: Figures 4–7 show that QuITE consistently improves predictive performance with generally modest additional cost. Except for S-Mamba, most backbones exhibit little to no increase in training or inference time. For fairness and reproducibility, we keep the original backbone hyperparameter settings from prior work [1]. In addition, as noted in Section 6.2, the full QuITE-based models are designed to have fewer parameters than their original backbone baselines. Therefore, **the observed gains are unlikely to be explained simply by larger model size.**
>
> [1] Luo, Yicheng, et al. "Hi-patch: Hierarchical patch gnn for irregular multivariate time series." *ICML*. 2025.
>
> ------
> **W4 / Q1. Why can a summarized representation outperform the baseline?**
>
> We agree that this is an important conceptual question. The relevant comparison is not **“full information” vs. “lossy summary,”** but rather **“irregularity-blind embedding” vs. “irregularity-aware adaptive summary.”** Without QuITE, MTS backbones cannot explicitly encode temporal irregularity.
>
> Consistent with this, replacing only the input embedding with QuITE already yields clear gains (Table 2), and even simple input-level adaptations such as **Add/Concat** improve performance (Table 5), suggesting that modeling irregularity at the input stage is particularly important. **QuITE provides the most effective version of this idea through learnable query-based aggregation.** This interpretation is further supported by the consistent gains in Table 2 and the t-SNE embeddings in Appendix G.

---

> > ### Author Rebuttal · Reviewer_QALz · 2026-04-01
> >
> > ### Complexity analysis
> >
> > I noticed line 377: authors claim that "QuITE-based models are designed with fewer parameters than their corresponding baselines". While there is **no evidence to support this claim**. First, those model with QuITE module are generally increase their complexity in terms of training/inference time. Second, the proposed QuITE++ model has no comparison in complexity to other baseline models. In the previous question, I have asked the author to provide empirical training/inference time or FLOPs complexity comparison, but no experimental data were provided.
> >
> > Although this may not be the most important concern, the authors' claims direct the reader to the concept: "Our method has few parameters and better performance". This claim needs to be supported with comprehensive evidence. FLOPs measure the total number of mathematical operations, while time measures how long it takes for hardware to perform those operations. train/inference time can be impacted by many factors, such as hardware efficiency and training configuration.
> >
> >
> >
> > ### Response of W4 / Q1
> >
> > The response is interesting. The statement **"modelling irregularity at the input stage is particularly important."** reminds me o another work [1] that has a similar intuition and experimental observation
> >
> > [1] Irregularity-informed time series analysis: Adaptive modelling of spatial and temporal dynamics
> >
> >
> > Other questions are well-addressed. Happy to raise the final score if all questions are well-addressed. Significance has been raised

---

> > > ### Author Response · Authors · 2026-04-05
> > >
> > > **Thank you for your thoughtful feedback.**
> > >
> > > We are encouraged to hear that the other concerns have been well addressed and that the significance score has been raised. We sincerely appreciate the opportunity to clarify the remaining issue regarding complexity.
> > >
> > > We agree that our original statement, **“QuITE-based models are designed with fewer parameters than their corresponding baselines,”** was not sufficiently supported in the previous response and could be misleading without a more comprehensive efficiency analysis. We also apologize for not providing the requested comparisons of FLOPs and training/inference time earlier.
> > >
> > > We do not claim that QuITE is universally more efficient in every respect. Rather, **our goal is to verify that the performance gains of QuITE are not simply a consequence of increased model capacity. In other words, even when additional runtime cost is incurred, the trade-off remains favorable given the improved predictive performance on IMTS.** To this end, following the reviewer’s suggestion in **W5 regarding complexity-controlled applicability**, we designed the QuITE-based variants to operate with fewer parameters than their corresponding original backbones.
> > >
> > > To substantiate this point, we now provide a comprehensive comparison across six representative forecasting backbones, along with QuITE++ and two SOTA IMTS baselines, reporting MSE, parameter count, FLOPs, and training/inference time per epoch (seconds) under identical settings. All experiments were conducted using the same hardware and training conditions (NVIDIA RTX A6000 GPU; Intel Xeon Gold 6426Y CPU, 16 cores), with the same batch size for each dataset (32 for Human Activity, 128 for USHCN, 64 for PhysioNet, and 8 for MIMIC-III) across multiple forecasting horizons.
> > >
> > > Due to the rebuttal space limit, we provide the detailed results in **Table 14 (overall averages)** and **Tables 15–16 (dataset-specific results)** at the following anonymous link:
> > >
> > > https://anonymous.4open.science/r/anonymous-review-images-1ED0
> > >
> > > From **Table 14** (where results are averaged across forecasting horizons for each dataset), we make three observations.
> > >
> > > - **First,** QuITE improves predictive performance across the reported settings **while using fewer parameters and lower FLOPs** than the corresponding backbones in our controlled comparisons. **This supports our main point that the gains do not arise from simply increasing model capacity, but from more effective modeling of irregularity.**
> > >
> > > - **Second,** runtime does not always correlate with parameter count or FLOPs. In some backbones, training and inference time decrease together with model size, while in others, runtime increases despite reduced parameters and FLOPs. This supports the reviewer’s point that runtime is influenced not only by model size, but also by architectural and implementation factors. **Even when additional runtime cost is observed, the trade-off remains favorable given the improved predictive performance on IMTS.**
> > >
> > > - **Third,** in comparison with strong IMTS baselines, QuITE++ achieves competitive or better accuracy on multiple datasets while maintaining a very small parameter budget. In terms of runtime, QuITE++ is consistently faster than Hi-Patch and faster than HyperIMTS on most datasets, although it is slower in some cases (e.g., PhysioNet). **Overall, QuITE++ shows competitive results among MSE, parameter count, FLOPs, and runtime relative to existing SOTA IMTS baselines.**
> > >
> > > In summary, we agree that our original wording should be made more precise. In the revision, we will revise the claim to avoid suggesting that QuITE is uniformly more efficient in all aspects. **Our intended conclusion is that QuITE’s gains do not come from increased model capacity, since the QuITE-based variants were deliberately constructed under a smaller parameter budget.** Rather, the evidence indicates that the gains stem from more effective modeling of irregularity. Even when additional runtime cost is observed, the trade-off remains favorable in light of the improved predictive performance on IMTS.
> > >
> > > As suggested by the reviewer, we will support this point with a comprehensive efficiency table reporting parameter count, FLOPs, and training/inference time separately.
> > >
> > > **We hope this clarification addresses your remaining concern, and we sincerely thank you again for your thoughtful and constructive feedback.**
> > >
> > > ---
> > > ---
> > > Dear Reviewer QALz,
> > >
> > > **Thank you very much for your positive feedback and for increasing your score.**
> > >
> > > We sincerely appreciate your thoughtful assessment and are encouraged to know that our rebuttal has resolved most of your concerns. To make the paper even stronger, we will carefully address your valuable comments in the revision.
> > >
> > > Best regards,
> > >
> > > The Authors

---

### Official Review · Reviewer_E6vp · 2026-03-12

**Soundness:** 3
**Presentation:** 2
**Significance:** 3
**Originality:** 3
**Overall Recommendation:** 4
**Confidence:** 5

**Summary:**

- This paper proposes a novel paradigm for Irregular Multivariate Time Series (IMTS) forecasting by replacing only the input embedding layer
- The core module QuITE generates fixed-dimensional latent representations by having a learnable query token set interact with irregular observations through a single self-attention layer

**Compliance With Llm Reviewing Policy:**

Affirmed.

**Key Questions For Authors:**

- Can you confirm whether QuITE's benefits stem from query-based irregular aggregation itself?
- Can you present a controlled experiment comparing it with alternative embeddings of the same parameter size?
- Where do the performance curve changes and the critical point where QuITE begins to fail occur?

**Limitations:**

- The failure conditions and cost limitations need to be described more explicitly.
- The sensitivity of QuITE's own hyperparameters was not adequately discussed in the main text.

**Strengths And Weaknesses:**

Strengths
- Well-designed controlled comparison where only QuITE's inclusion/exclusion varies while keeping everything else identical
- An approach enabling the reuse of existing, validated MTS models for IMTS
- Persuasive perspective shift redefining the IMTS problem as a limitation of the embedding layer

Weaknesses
- Difficulty distinguishing how performance improvements are achieved in detail.
- Little analysis of failure cases or conditions where improvements are minimal.
- Unpredictability of when QuITE reaches its limits.

---

> ### Author Rebuttal · Authors · 2026-03-31
>
> **W1 / Q1. How are the improvements achieved; do gains stem from query-based aggregation?**
>
> We believe the gains primarily stem from query-based aggregation itself.
> - **QuITE explicitly models irregularity at the input stage, which MTS models are not designed to handle.** Table 5 and Appendix E show that even simple input-level adaptations such as Add/Concat already improve performance, indicating that encoding irregularity is beneficial.
> - **QuITE handles this most effectively through learnable query-based aggregation.** Because IMTS are asynchronous and variable-length, direct attention yields observation-level representations that are not directly compatible with MTS backbones, while simple pooling may dilute irregularity information. **QuITE instead uses learnable query tokens for adaptive, timestamp-aware aggregation, producing structured embeddings without artificial values.** This is supported by the stronger results in Tables 2 and 5, as well as the t-SNE embeddings in Appendix G.
>
> ---
> **Q2 : Can you present a controlled experiment comparing it with alternative embeddings of the same parameter size?**
>
> Yes. This comparison is already included in Table 5. In particular, Mean serves as a controlled comparison because it removes only the query tokens while keeping the rest of the embedding pipeline unchanged, thereby isolating the effect of learnable query-based aggregation. mTAND also has a parameter size similar to QuITE, making it a relevant alternative of comparable scale. Since QuITE consistently outperforms these baselines, the results suggest that its advantage comes not merely from model size, but from its embedding design. We will include the detailed parameter counts in the revision to make this point more explicit.
>
> ---
> **W2 / W3 / Q3 / L1. Failure cases and limits.**
>
> We agree that the failure conditions and limits of QuITE should be described more explicitly. To better analyze this, we conducted an additional experiment on QuITE++ in which observations were randomly removed at different rates and forecasting performance was measured.
>
> The results show a gradual degradation as the missing ratio increases. Performance remains relatively stable up to about 50% additional removal, suggesting that QuITE can robustly handle sparse IMTS, which is also consistent with the nature of IMTS. However, when the removal rate reaches **75%**, the performance drop becomes much larger, indicating a practical **critical point** beyond which the available observations become too sparse for reliable forecasting. we will include this discussion in the revision.
>
> | DataSet                     | Metric | 0%   | 25%  | 50%  | 75%  |
> | --------------------------- | ------ | ---- | ---- | ---- | ---- |
> | Activity (3000ms →  1000ms) | MSE    | 2.46 | 2.48 | 2.48 | 2.60 |
> |                             | MAE    | 2.92 | 3.02 | 3.04 | 3.11 |
> | USHCN (24m →  12m)          | MSE    | 4.81 | 4.85 | 4.82 | 4.90 |
> |                             | MAE    | 2.93 | 2.95 | 2.93 | 2.95 |
>
> ---
> **L2. Sensitivity of QuITE’s own hyperparameters.**
>
> We agree that the sensitivity of QuITE’s own hyperparameters should be discussed more clearly. For QuITE applied to MTS backbones, the hyperparameters are fixed to nhidms = 64 and nheads = 4, so no separate sensitivity study was conducted in that setting. For QuITE++, we analyzed nhidms, nlayers, and nheads using horizon-averaged MSE/MAE, and found that performance is generally **robust to these choices**.
>
> |  Dataset  | nhdims | MSE  | MAE  | nlayers | MSE  | MAE  | nheads | MSE  | MAE  |
> | :-------: | :----: | :--: | :--: | :-----: | :--: | :--: | :----: | :--: | :--: |
> | Activity  |   32   | 3.22 | 3.55 |    1    | 3.19 | 3.58 |   1    | 3.26 | 3.58 |
> |           |   64   | 3.22 | 3.54 |    2    | 3.07 | 3.44 |   2    | 3.23 | 3.61 |
> |           |        |      |      |    3    | 2.91 | 3.34 |   4    | 3.07 | 3.46 |
> |           |        |      |      |         |      |      |   8    | 2.91 | 3.32 |
> |   USHCN   |   32   | 4.86 | 2.97 |    1    | 4.86 | 2.97 |   1    | 4.91 | 3.02 |
> |           |   64   | 4.84 | 2.96 |    2    | 4.85 | 2.97 |   2    | 4.85 | 2.96 |
> |           |        |      |      |    3    | 4.85 | 2.95 |   4    | 4.83 | 2.95 |
> |           |        |      |      |         |      |      |   8    | 4.83 | 3.00 |
> | PhysioNet |   32   | 5.04 | 3.61 |    1    | 4.45 | 3.42 |   1    | 5.10 | 3.67 |
> |           |   64   | 5.07 | 3.64 |    2    | 4.76 | 3.63 |   2    | 4.86 | 3.85 |
> |           |        |      |      |    3    | 4.90 | 3.77 |   4    | 4.68 | 3.49 |
> |           |        |      |      |         |      |      |   8    | 4.94 | 3.62 |
> | MIMIC-III |   32   | 1.67 | 7.07 |    1    | 1.60 | 6.89 |   1    | 1.68 | 7.16 |
> |           |   64   | 1.70 | 7.24 |    2    | 1.68 | 7.07 |   2    | 1.66 | 7.06 |
> |           |        |      |      |    3    | 1.72 | 7.25 |   4    | 1.68 | 7.09 |
> |           |        |      |      |         |      |      |   8    | 1.68 | 7.07 |

---

> > ### Author Rebuttal · Reviewer_E6vp · 2026-04-03
> >
> > The rebuttal addresses most of my main concerns. In particular, the authors provided a clearer explanation of why the gains are attributed to query-based irregular aggregation, added a more concrete analysis of failure conditions through the missing-ratio experiment, and clarified the sensitivity of the method’s own hyperparameters.
> > I find these additions helpful and relevant. The missing-ratio results are especially useful because they make the practical limitation of the method more explicit, rather than leaving it only as a general statement. The clarification on controlled comparisons with alternative embeddings also improves my confidence that the gains are not simply due to parameter count.
> > Some points, such as cost/limitation discussion, should still be stated more explicitly in the revised manuscript, but overall the rebuttal sufficiently addresses my main questions. I therefore maintain my positive overall assessment.

---

> > > ### Author Response · Authors · 2026-04-06
> > >
> > > Dear Reviewer E6vp,
> > >
> > > **Thank you very much for your thoughtful and encouraging feedback. We sincerely appreciate your detailed evaluation and are glad that our rebuttal has addressed most of your main concerns.**
> > >
> > > We are particularly grateful for your recognition of the additional analyses, including the clarification of the gains from query-based irregular aggregation, the missing-ratio experiments, and the discussion on hyperparameter sensitivity and controlled comparisons. It is encouraging to hear that these additions improved the clarity and strengthened your confidence in our work.
> > >
> > > We also appreciate your suggestion to further elaborate on the cost and limitations. In the revised manuscript, we will make these aspects more explicit to provide a clearer and more balanced presentation.
> > >
> > > Thank you again for your valuable feedback and for maintaining a positive overall assessment of our work.
> > >
> > > Best regards,
> > >
> > > The Authors

---

### Official Review · Reviewer_5Qm5 · 2026-03-13

**Soundness:** 2
**Presentation:** 2
**Significance:** 3
**Originality:** 3
**Overall Recommendation:** 4
**Confidence:** 4

**Summary:**

This paper proposes QuITE (Query-based Irregular Time-series Embedding), a backbone-agnostic, plug-and-play input embedding module designed to enable existing multivariate time series (MTS) forecasting models to directly process irregular multivariate time series (IMTS). The core idea is attention-based aggregation mechanism driven by learnable query tokens. Each query token aggregates a subset of irregular observations via a single self-attention layer, producing fixed-dimensional latent representations that are compatible with any downstream MTS backbone. The authors further introduce QuITE++, a hierarchical encoder that stacks patch-level and variable-level self-attention layers, combined with a time-query-based cross-attention decoder for arbitrary-timestamp prediction. Experiments are conducted on four real-world IMTS benchmarks (Human Activity, USHCN, PhysioNet, MIMIC-III), demonstrating consistent performance improvements across six different backbone architectures and comparisons against seventeen baselines.

**Compliance With Llm Reviewing Policy:**

Affirmed.

**Final Justification:**

This paper proposes QuITE (Query-based Irregular Time-series Embedding), a backbone-agnostic, plug-and-play input embedding module designed to enable existing multivariate time series (MTS) forecasting models to directly process irregular multivariate time series (IMTS). The core idea of exploiting the set operation property of attention for irregular time series modeling is novel and compelling. While additional analysis regarding the method's compatibility with diverse model architectures would strengthen the work, the results presented are already solid.

**Key Questions For Authors:**

See weaknesses

**Limitations:**

see Weaknesses

**Strengths And Weaknesses:**

## Strength

- The problem framing is well-motivated. The authors correctly identify that the bottleneck for applying standard MTS models to IMTS lies in the input embedding stage, which is a non-obvious and insightful observation.
- The proposed attention-based aggregation with learnable query tokens is a principled mechanism. The connection to the [CLS] token in BERT is a reasonable analogy and clearly explained.
- The plug-and-play nature of QuITE is a practical strength: it allows practitioners to leverage advances in MTS modeling without redesigning architectures for irregular data.
- The ablation in Table 5, comparing Add, Concat, mTAND, Mean pooling, and QuITE, is well-designed and effectively isolates the contribution of the learnable query tokens over simpler aggregation alternatives.

## Weaknesses
- A significant and concerning result is that patch-based backbones (PatchTST, PatchMixer, TMix) with QuITE perform dramatically worse than IMTS-specific baselines on PhysioNet and MIMIC-III. The authors acknowledge this in passing but do not provide a satisfying explanation. This suggests that QuITE's effectiveness is highly architecture-dependent, which limits the claim of being universally beneficial for all MTS backbones. The authors should analyze why patch-based models fail in these high-missingness scenarios even with QuITE, and whether there are principled remedies.
- Missing ablation on the number of query tokens: No ablation is provided on varying this number or on alternative query initialization strategies.
- While QuITE is shown to improve variate-token and hybrid models substantially, it does not consistently bring patch-based models to competitive performance across all datasets. A more nuanced characterization of when QuITE is most beneficial is needed.
- QuITE and QuITE++ are not cleanly separated in the narrative, making it slightly unclear whether the main contribution is the embedding module or the new hierarchical model,since the main contribution claimed is the former.
- "typos/inconsistencies such as “USCHN” vs “USHCN”, “QUIET” vs “QuITE”, “Wrapformer” vs “Warpformer”, “to MTS models to MTS” in Sec. 6.3. Typo in Table 4 caption: "dastset" should be "dataset.

---

> ### Author Rebuttal · Authors · 2026-03-30
>
> **W1 / W3. Patch-based backbones underperform on some datasets; a more nuanced characterization of when QuITE is most beneficial is needed.**
>
> Thank you for this important comment. We agree that the paper should more clearly characterize **when QuITE is most effective,** especially since QuITE-equipped patch-based backbones remain substantially weaker than several IMTS-specific baselines on PhysioNet and MIMIC-III.
>
> We also agree that the current results show an **architecture-dependent pattern in the final gains**. However, we would like to clarify that this pattern should be understood primarily in terms of the **backbone’s ability to exploit the adapted representation**, rather than as evidence that QuITE itself benefits only a narrow class of models.
>
> QuITE is best viewed as an **input-adaptation module** whose role is to transform IMTS into structured embeddings so that existing MTS backbones can process IMTS directly. From this perspective, its broad applicability is supported by the **consistent relative improvements** observed across datasets, backbone families, and forecasting horizons.
>
> QuITE’s effectiveness therefore depends on (1) how sensitive a backbone is to irregular sampling and (2) how well it can leverage the adapted embeddings.
>
> - First, consistent with Table 2, variate-token models show larger gains than patch-based models, likely because they **summarize entire sequences** and are therefore more strongly affected by irregularity in the input.
> - Second, this distinction is particularly important for interpreting the patch-based results on PhysioNet and MIMIC-III. The weaker performance of PatchTST, PatchMixer, and TMix appears to reflect their largely **variable-independent modeling bias**, which can be especially limiting in **clinical IMTS**, where **cross-variable interactions** are likely crucial [1]. We believe this interpretation is further supported by **TimeXer**: although it shares the same patch-based foundation as PatchTST, it additionally incorporates explicit cross-variable modeling and performs substantially better on these datasets.
>
> We will revise the paper accordingly to clarify that QuITE is broadly applicable as an **input-adaptation interface**, while its ultimate benefit depends on whether the downstream backbone can effectively exploit the structured embeddings it provides.
>
> [1] Raindrop: Graph-Guided Network for Irregularly Sampled Multivariate Time Series (ICLR 2022)
>
> ---
> **W2. Missing ablation on the number of query tokens / initialization.**
>
> - The **number of query tokens** is not a hyperparameter; it is determined by the backbone-compatible target structure (Section 4.2), i.e., per variable or per patch-variable pair.
>
> - For query initialization, we currently use standard random initialization. We have additionally conducted experiments with alternative initialization schemes (e.g., Xavier, uniform, and zero) and added the corresponding table. The results indicate that **QuITE is robust to the choice of initialization, with only minor performance differences across schemes.**
>
> | DataSet                     | Metric | Xavier | Uniform | Zero | Random |
> | --------------------------- | ----- | ------ | ------ | ---- | ------- |
> | Activity (3000ms → 1000ms) | MSE   | 2.46   | 2.45   | 2.46 | 2.46    |
> |                             | MAE   | 3.00   | 2.99   | 3.01 | 2.92    |
> | USHCN (24m → 12m)          | MSE   | 4.83   | 4.86   | 4.87 | 4.81    |
> |                             | MAE   | 2.99   | 2.95   | 2.98 | 2.93    |
>
> ---
> **W4. Separation between QuITE and QuITE++.**
>
> We agree that this distinction should be made clearer. The **main contribution of the paper is QuITE**, namely an **input-embedding module** that converts IMTS into structured representations that standard MTS backbones can directly use, without data distortion. In contrast, **QuITE++ is not the primary contribution, but an extension built on top of this core idea**.
>
> More specifically, the key contribution of the paper is the **embedding principle** itself: using learnable query tokens to transform IMTS into structured representations. **QuITE++ serves to show that this same principle can be extended beyond input embedding into a hierarchical modeling framework.**
>
> - It first uses query tokens to form **patch-level representations** from IMTS.
> - It then introduces **variable-level summary tokens** to construct higher-level variable representations.
>
> In this sense, QuITE++ should be understood as an example of how the core idea behind QuITE can be naturally extended to hierarchical modeling, rather than as a separate main contribution.
>
> We will revise the manuscript to make this distinction explicit and to clearly state that **QuITE is the main contribution, while QuITE++ is an extension built upon it**.
>
> ---
> **W5. Typos / inconsistencies.**
>
> We thank the reviewer for the careful reading. We will correct these typos and inconsistencies in the revision.

---

> > ### Author Rebuttal · Reviewer_5Qm5 · 2026-04-05
> >
> > I appreciate the authors' thorough response and the new experimental results, which have substantially addressed my concerns. I am happy to raise my score.

---

> > > ### Author Response · Authors · 2026-04-06
> > >
> > > Dear Reviewer 5Qm5,
> > >
> > > **Thank you very much for your encouraging feedback, your positive assessment of our work, and for raising your score.**
> > >
> > > We are delighted to hear that our rebuttal has effectively addressed your main concerns. We will carefully reflect all of your comments in the revision to further strengthen the paper. We also sincerely appreciate your support and favorable evaluation.
> > >
> > > Best regards,
> > >
> > > The Authors

---

### Official Review · Reviewer_zUZ8 · 2026-03-13

**Soundness:** 3
**Presentation:** 3
**Significance:** 3
**Originality:** 2
**Overall Recommendation:** 4
**Confidence:** 3

**Summary:**

This paper studies the problem of Irregular Multivariate Time Series (IMTS) modeling. Due to characteristics such as irregular sampling and asynchronous observations, existing approaches typically either rely on irregularity-specific architectures or convert data into regular time series through interpolation or imputation. However, these approaches either limit the reuse of existing Multivariate Time Series (MTS) models or may introduce artificial values that distort the true temporal dynamics.

To address these issues, the paper proposes an input-embedding-based approach, where irregular observations are handled at the embedding layer, while keeping existing MTS backbone architectures unchanged. Based on this idea, the authors introduce QuITE (Query-based Irregular Time-series Embedding). QuITE employs learnable query tokens and self-attention aggregation to aggregate irregular observations and produce fixed-dimensional representations, enabling standard MTS models to directly process IMTS data. The module is backbone-agnostic and plug-and-play, allowing it to be easily integrated into various models.

Furthermore, the authors propose QuITE++, a hierarchical encoder that models both temporal dependencies and cross-variable interactions through patch-level attention and variable-level attention, respectively. A time-query-based decoder is then used for forecasting. Experiments conducted on several IMTS benchmarks demonstrate the effectiveness of the proposed approach, showing that QuITE consistently improves the forecasting performance of multiple MTS backbones.

**Compliance With Llm Reviewing Policy:**

Affirmed.

**Final Justification:**

The authors have addressed my concerns and I will maintain my score.

**Key Questions For Authors:**

1. The experimental results show substantial performance improvements across several datasets. Could the authors provide further insights into the main factors driving these improvements?

2. QuITE uses self-attention aggregation to model irregular observations. When the number of observations becomes large, the computational complexity may increase. How does this affect scalability in practice?

3. The forecasting stage adopts a time-query-based decoder with cross-attention. Why was a cross-attention decoder chosen instead of a simpler MLP-based decoder?

**Limitations:**

The paper currently does not explicitly discuss limitations. It is recommended to add a dedicated section addressing the following aspects:

1. Scope and limitations of the proposed method.
   Since QuITE relies on attention-based aggregation to process irregular observations, the computational complexity may increase when the number of observations becomes large.

2. Scalability for large-scale time-series data.
   The authors could discuss the applicability of the method to large-scale datasets, as well as possible optimization strategies to mitigate computational overhead.

**Strengths And Weaknesses:**

### Strengths

1. Methodological novelty with good generality and scalability.
   The proposed QuITE models irregular observations directly at the embedding layer through learnable query tokens combined with attention aggregation, enabling existing MTS backbone models to process IMTS data without structural modifications. This design has strong practical potential due to its generalizability and extensibility.

2. Comprehensive experimental evaluation.
   The method is evaluated on several real-world IMTS benchmark datasets, including Human Activity, USHCN, PhysioNet, and MIMIC-III. The experiments compare the approach with both IMTS-specific models and standard MTS forecasting models, providing a relatively comprehensive experimental setup.

3. Strong empirical performance.
   Across multiple backbone architectures and different prediction horizons, QuITE consistently delivers notable performance improvements.

---

### Weaknesses

1. The proposed QuITE (Query-based Irregular Time-series Embedding) aggregates irregular observations using learnable query tokens and self-attention. Conceptually, this design shares similarities with the [CLS] token mechanism in BERT and other attention-based pooling methods used for representation aggregation.
   The authors are encouraged to clarify the relationship and differences between QuITE and these existing approaches.

2. The paper claims that *“the primary limitation lies not in the backbone architecture itself”*, suggesting that the difficulty in modeling irregular time series mainly originates from the input embedding layer rather than the backbone architecture. However, the current justification appears somewhat intuitive.
   It would be beneficial to further explain:
   - Why query token aggregation is particularly effective for handling irregular sampling in IMTS.
   - Why improving the embedding layer is more critical than modifying the backbone architecture.

3. It is recommended that the authors release the code and detailed evaluation procedures to facilitate reproducibility and enable fair comparisons with other approaches, which would also help advance research in this area.

4. The paper reports performance improvements ranging from 3.5% to 45.9% across different models and datasets, which is a relatively wide range. It would be helpful to analyze:
   - Why improvements are smaller on certain backbones or datasets but significantly larger on others.
   - Whether these differences are related to factors such as the degree of irregularity, sequence length, or dataset characteristics.

5. The current ablation studies are relatively limited. More systematic ablation experiments could be included, such as:
   - The impact of the number of query tokens.
   - The effect of different aggregation designs.
   These analyses would help better demonstrate the contributions of each component in QuITE.

---

> ### Author Rebuttal · Authors · 2026-03-30
>
> We thank the reviewer for the helpful comments.
>
> ---
> **W1. Relation to [CLS] / attention pooling.**
>
> - As noted in the introduction, QuITE is conceptually related to the [CLS] token, but our contribution is that **its reinterpretation as an input-embedding module for IMTS**.
> - Unlike generic attention pooling, QuITE converts IMTS into the **structured inputs required by MTS backbones** (e.g., $[B,N,D]$, $[B,M,N,D]$). Direct self-attention over IMTS yields observation-level representations (e.g., $[B,N,L,D]$) that are not directly compatible with MTS backbones, while simple pooling may dilute temporal irregularity information. QuITE instead **uses learnable query tokens for adaptive aggregation** to produce backbone-compatible structured embeddings.
>
> ---
> **W2-1. Why query aggregation is effective.**
>
> IMTS are asynchronous and vary in length across variables; thus, uniform aggregation (e.g., attention mean-pooling) can wash out informative observations. **QuITE leverages learnable query tokens to perform adaptive, timestamp-aware aggregation, enabling each variable or patch-variable unit to selectively summarize relevant observations without data distortion.** This is consistent with the results in Table 5, where QuITE outperforms existing methods.
>
> ---
> **W2-2. Why the embedding layer matters more.**
>
> **Since MTS models assume regular sampling, so IMTS primarily creates a mismatch at the input representation stage**. In this sense, the issue resembles **“garbage in, garbage out.”** Replacing only the input embedding with QuITE already yields consistent gains (Table 2), and even simple input-level adaptations such as Add/Concat improve performance (Table 5), suggesting that encoding irregularity at the input level is especially critical. Appendix G further supports this through more compact and separable t-SNE embeddings.
>
> ---
> **W3. Reproducibility.**
>
> We will further improve the documentation and release the code to facilitate reproducibility.
>
> ---
> **W4. Why gains vary across models/datasets.**
>
> - QuITE is best viewed as an **input-adaptation module**, not a generic performance booster. Accordingly, its gains depend on how sensitive each backbone is to irregular sampling. Consistent with Table 2, variate-token models show larger gains than patch-based models, likely because they **summarize entire sequences** and are therefore more affected by irregularity.
> - On PhysioNet and MIMIC-III, the weaker performance of patch-based backbones appears to reflect their **variable-independent modeling bias**, rather than a limitation of QuITE itself. This can be especially limiting for clinical datasets, where cross-variable interactions are likely important [1]. This interpretation is further supported by TimeXer. Its stronger performance shows that **adding cross-variable modeling on top of a patch-based architecture (same as PatchTST) leads to clear gains**.
>
> [1] Raindrop: Graph-Guided Network for Irregularly Sampled Multivariate Time Series (ICLR 2022)
>
> ---
> **W5. Ablation.**
>
> - The **# of query tokens** is not a hyperparameter; it is determined by the backbone-compatible target structure (Section 4.2), i.e., per variable or per patch-variable pair.
> - For the **aggregation design**, Table 5 compares QuITE with Add, Concat, mTAND, and Mean, and QuITE consistently performs best. Unlike mTAND, QuITE operates directly on IMTS, while its gains over Mean indicate the importance of **adaptive query-based aggregation** beyond self-attention alone (Appendix E.2).
>
> ---
> **Q1. Main factors driving improvement.**
>
> The main factor is that **QuITE explicitly models irregularity at the input stage, which standard MTS models are not designed to handle.** Table 5 and Appendix E show that even simple input-level adaptations already improve performance, indicating that properly encoding irregularity is beneficial. **QuITE provides the most effective version of this idea through learnable query-based aggregation.** This is further supported by the consistent gains in Table 2 and the t-SNE embeddings in Appendix G.
>
> ---
> **Q2 / L1 / L2. Scope / limitations / scalability.**
>
> **Since IMTS is typically sparse**, the practical overhead of QuITE is limited. **As attention is added only at the embedding stage**, the overall cost remains moderate relative to the gains (Appendix F). However, for very dense or long sequences, additional optimization such as sparse attention may be needed.
>
> ---
> **Q3. Why cross-attention decoder.**
>
> We chose a cross-attention decoder to provide a fair evaluation across backbone types. Unlike an MLP decoder, which requires compressing features into a single vector (e.g., $[B,N,D]$) and introduces extra pooling for patch-based models (e.g., $[B,M,N,D]$). **cross-attention allows both variable-based and patch-based models to condition predictions directly on future time queries under the same decoding setup, especially appropriate for IMTS forecasting at arbitrary future timestamps**.

---

> > ### Author Rebuttal · Reviewer_zUZ8 · 2026-04-03
> >
> > Thank you for the response. I will maintain my current score.

---

> > > ### Author Response · Authors · 2026-04-06
> > >
> > > Dear Reviewer zUZ8,
> > >
> > > **We are pleased that our rebuttal has helped address your questions and comments.**
> > >
> > > We sincerely appreciate the time and effort you devoted to reviewing our paper and for providing such detailed and constructive feedback, which has helped guide the paper toward a more constructive and improved direction. We will make sure to carefully incorporate all of your comments in the revision to further enhance the quality of the paper. We are also grateful for your continued positive evaluation.
> > >
> > > Best regards,
> > >
> > > The Authors

---

### Decision · Program_Chairs · 2026-04-30

**Decision:**

Accept (regular)

**Comment:**

The four reviewers agreed on Weak Accept (4). They consistently find the paper well-motivated, with a clear and practical perspective shift that attributes the core difficulty of IMTS modeling to the input representation rather than the backbone. The proposed query-based aggregation mechanism is considered technically sound and reasonably novel, and the plug-and-play design is viewed as a practically valuable contribution. Empirical results across multiple datasets and backbones demonstrate consistent performance improvements, further supporting the effectiveness of the approach.

The rebuttal is effective and successfully addresses most reviewer concerns. As a result, all reviewers acknowledge that their concerns have been fully or largely resolved, and some explicitly increase their confidence or scores.

Overall, I am happy to recommend accept.